# Integrating protein language and geometric deep learning models for enhanced vaccine antigen prediction

Xiaodong Zai [1,3], Yunxiang Zhao [1,3], Xiaolin Wang[1], Mingyue Leng[2], Menglong Lu[2], Yilong Yang [1], Xiaofan Zhao[1], Ruihua Li [1], Yaohui Li[1], Yue Zhang[1], Jun Zhang [1], Dongsheng Li[2], Hongguang Ren [1] ✉, Junjie Xu [1] ✉ & Wei Chen [1] ✉

Vaccines are the most effective tool in preventing and managing infectious diseases. One of the critical challenges in vaccine development is the selection of suitable target antigens from the thousands of proteins produced by pathogens. Artificial intelligence is anticipated to play a significant role in addressing this challenge. In this study, we develop a framework termed PLGDL for protective antigen prediction that employs Protein Language and Geometric Deep Learning models. This framework leverages both primary sequence features and three-dimensional structural features of protein antigens, thereby reducing the biases associated with manually curated features. Our integrated model exhibits robustness across both constructed and public datasets and is applicable to viruses, bacteria, and eukaryotic pathogens. Notably, when applied to the ongoing Mpox outbreak, our model not only quickly identifies multiple known antigens but also discovers a protective antigen: G10R. Here, our study provides a high-performance screening tool for protective vaccine antigen prediction by synergistically utilizing the capabilities of protein language and geometric deep learning models, providing substantive insights and methodological advancements for rapid vaccine development.

Vaccination is the cornerstone of global public health since it offers the most effective means to prevent and control infectious diseases[1]. The precise discovery of candidate antigens, which elicit specific protective immune responses, is fundamental to successful vaccine development[2]. Artificial intelligence has already led to incredible breakthroughs in various domains: from the game of Go to self-driving vehicles, to formal mathematical proofs[3–6]. It is also poised to play a significant role in the fight against infectious diseases, particularly in vaccine development[7–10]. Reverse vaccinology (RV), which leverages bioinformatics and machine learning approaches to systematically predict and identify candidate antigens from pathogen genomes, has

emerged as a powerful filtering and prioritization tool that directs limited experimental resources toward the most promising candidates, thereby improving the overall efficiency of vaccine development[11,12].

Previous research has shown that the probability of proteins being vaccine antigens is strongly correlated with their biological and physicochemical properties[13]. Accordingly, existing machine learning RV approaches, such as the widely used VaxiJen and Vaxign-ML models, have relied on manually engineered features derived from protein sequences. These features include physicochemical properties (like molecular weight, isoelectric point, and hydrophobicity) and

---

[1]Laboratory of Advanced Biotechnology, Beijing Institute of Biotechnology, Beijing, China. [2]College of Computer, National University of Defence Technology, Changsha, China. [3]These authors contributed equally: Xiaodong Zai, Yunxiang Zhao. ✉e-mail: bioren@163.com; xujunjie@sina.com; cw0226@foxmail.com

biological characteristics (like subcellular localization, adhesin probability, transmembrane helices, and signal peptides)[14,15]. While these handcrafted features demonstrate utility, they are inherently constrained by prior knowledge and may fail to capture more complex, multifaceted relations between sequence and antigenicity. The advent of deep learning models, which can automatically learn hierarchical feature representations from raw data, offers a promising alternative[16]. Such models can uncover intricate patterns and dependencies that are not readily apparent through traditional feature engineering. This shift from manual to automated feature extraction represents a major advancement in the field of RV, enabling more accurate and robust predictions of vaccine candidates, as evidenced by recent models like Vaxign-DL and Vaxi-DL[17,18].

In recent years, it has been shown that features extracted from pre-trained, task-independent protein language models can greatly improve classification performance in many biological problems[19–22]. These models, often based on architectures such as transformers, are pre-trained on large datasets of protein sequences, enabling them to learn intricate patterns and dependencies inherent to the sequences. This pretraining equips the models with a deep understanding of the biochemical properties and evolutionary relationships that govern protein function[23]. One of the key advantages of protein language models is their ability to capture long-range dependencies and complex interactions within protein sequences. Traditional sequence-based features are generally limited by their reliance on predefined characteristics. In contrast, protein language models can autonomously derive high-dimensional feature representations that encapsulate both local and global sequence information[24]. By capturing the contextual and semantic information encoded in protein sequences, these models have the potential to improve antigen prediction beyond what can be achieved with traditional sequence-based features.

Furthermore, a protein's structure is a critical determinant of its antigenic properties because the spatial arrangement of amino acids governs the accessibility and presentation of epitopes to the immune system[25,26]. However, current antigen prediction models primarily focus on one-dimensional (1D) sequence features and thus potentially neglect valuable structural information. This holistic approach, combining sequence and structure-based features, may offer a robust framework for advancing the field of RV. The recent progress in protein structure prediction, as exemplified by AlphaFold2 and AlphaFold3, provides an opportunity to extract structural features in an automated fashion[27,28]. This advancement opens new avenues for extracting structural features that were previously inaccessible. Although manually designing appropriate structural features is inherently difficult because of the complexity of protein folding, geometric deep learning models, which extend traditional deep learning methods to non-Euclidean spaces such as graphs and manifolds, have made significant strides in representing the three-dimensional (3D) characteristics of proteins[29–33].

In the present study, we propose a framework called PLGDL, as shown in Fig. 1. It simultaneously leverages protein language and geometric deep learning models to integrate both sequence features and structural features for predicting protective vaccine antigens. On both constructed and public datasets, PLGDL outperforms traditional methods that rely solely on primary sequence features. Moreover, in response to the recent Mpox outbreak, declared a "Public Health Emergency of International Concern" by the WHO. Our model not only identifies multiple known antigens but also discovers a previously uncharacterized antigen, G10R, which shows neutralizing and partial protective efficacy against a lethal orthopoxvirus challenge. This finding highlights G10R's potential as a component of new Mpox vaccines and supports ongoing vaccine development efforts.

Our study presents a high-performance protective antigen prediction model by leveraging the combined strengths of protein language and geometric deep learning models. This integrated RV approach not only provides substantive insights but also represents a significant advancement in vaccine development. The practical value of our model is exemplified by its successful application to the Mpox virus, underscoring its potential to rapidly respond to emerging infectious disease threats.

## Results
### Establishment and evaluation of the antigen dataset
To develop our machine learning framework for vaccine antigen prediction, it was essential to first establish a comprehensive and diverse dataset of positive and negative protective antigens encompassing various typical pathogen types. In this study, we exclusively included protective antigens defined as proteins that meet the following criteria: demonstrated ability to induce protection against pathogen challenge in vivo or capacity to stimulate antigen-specific immune responses directly linked to protective outcomes[34]. This stringent criterion, applied to data from the Protegen Database and PubMed-curated studies, yielded a high-confidence set of 600 protective antigens after removing redundancies (composition: Virus 119, Bacteria 386, Eukaryota 95; Fig. 2a; Supplementary Fig. 1).

To reflect the natural scarcity of protective antigens, we constructed an imbalanced negative dataset by extracting all annotated protein sequences from the genomes of the pathogens listed in the UniProt database, randomly selecting and excluding homologous sequences to positive antigens[35]. The final curated dataset contains 6000 negative samples after removing redundancies (composition: Viral 481, Bacterial 4493, Eukaryotic 1026; Fig. 2a and Supplementary Fig. 1), establishing a 1:10 positive-negative ratio that mirrors biological reality. The antigen dataset encompasses a broad spectrum of sequence lengths (ranging from 60 to 1923 amino acids), ensuring its broad applicability across diverse biological contexts (Fig. 2b and Supplementary Data 1).

All 6600 protein structures were reanalyzed using AlphaFold3 (Fig. 2c)[28]. Structural confidence was evaluated using key metrics—pLDDT (Predicted Local Distance Difference Test), pTM (Predicted TM-Score), and Predicted Aligned Error (PAE), resulting in high-quality predictions (mean pLDDT = 81.69). Analyses revealed a weak negative correlation between pLDDT and protein length ($R^2 = 0.0135$, Fig. 2d), confirming consistent prediction quality across antigen sizes. Notably, pTM (mean = 0.731) and PAE (mean = 9.925 Å) exhibited a strong inverse correlation ($R^2 = 0.8137$, Fig. 2e), indicating that antigens with higher global structural confidence (pTM) display lower inter-domain positional uncertainty (PAE). This dataset represents a comprehensive structural resource for protective antigens, now publicly available to advance vaccine research (Supplementary Data 2).

### Establishment of the PLGDL framework based on integrating sequence and structural features
Through systematic evaluation of state-of-the-art protein embedding methods, including ESM-2[23], ProTrans[36], and AMPLIFY[37], we identified ESM-2 as the optimal choice for antigen prediction tasks (Supplementary Table 1). Consequently, ESM-2[23] was adopted to generate residue-level feature embeddings from antigen sequences. These high-dimensional embeddings were subsequently refined through several feature selection pipelines, reducing dimensionality to 255 discriminative features, which were ultimately utilized as input for our antigen prediction framework (Supplementary Fig. 2 and Supplementary Table 2). The 3D structure of a protein ultimately determines its biological characteristics and physicochemical parameters, and it contains far more information than a 1D amino acid sequence. Compared to sequence features, protein structural features may be more indicative of the innate properties of antigens[25,26]. Therefore, the neighborhood-enhanced graph convolutional network (NEGCN) was established to accurately characterize protein structures and extract relevant feature vectors. To achieve this, we obtained a substantial

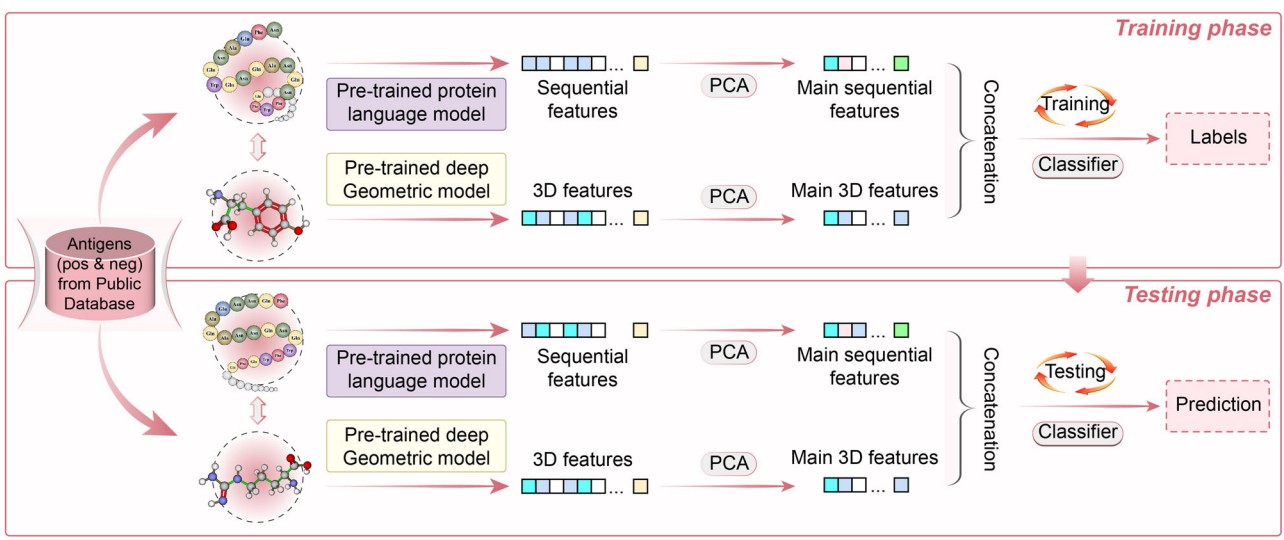

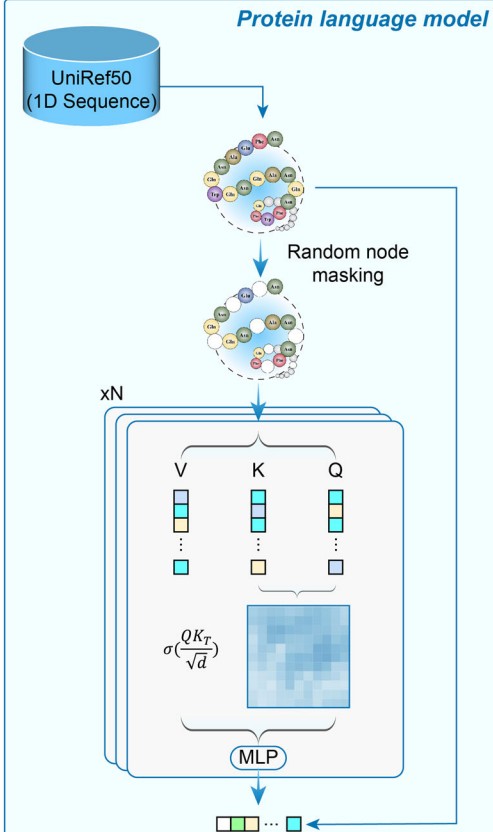

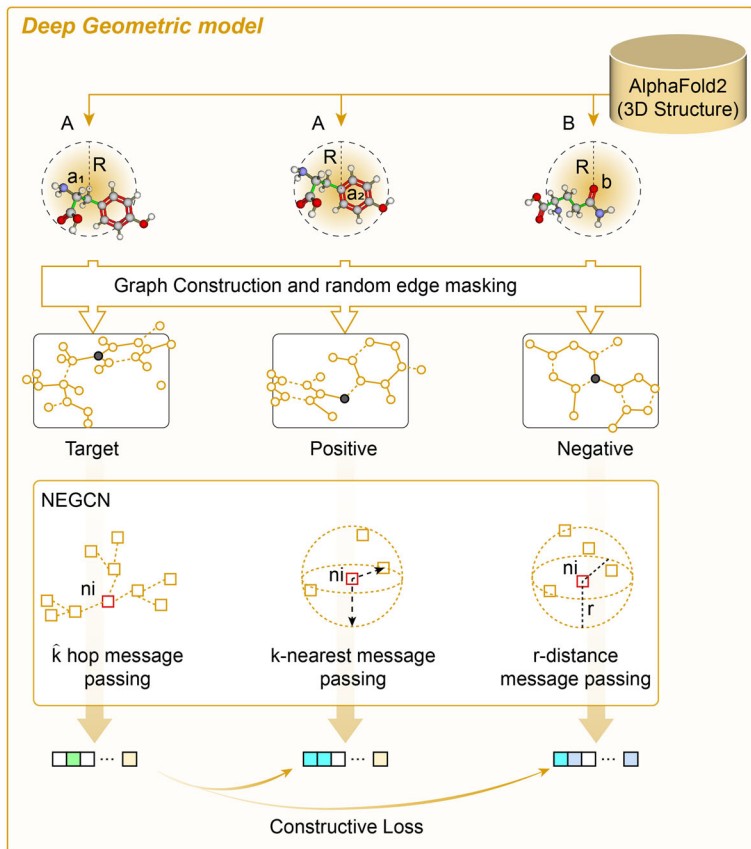

**Fig. 1 | The PLGDL framework for vaccine antigen prediction.** The first step in the framework involves the establishment and evaluation of a comprehensive protective antigen dataset. Next, a pre-trained protein language model was first used for protein sequence feature extraction. Considering the richness of information in protein structures, an improved neighbor-enhanced graph convolutional network (NEGCN) model was used to represent protein structures, and the extracted features were combined with the extracted amino acid sequence features. The resulting machine learning classifier was finally used in protective vaccine antigen prediction.

dataset of 805,000 high-quality protein structures from the AlphaFold Protein Structure Database[38]. This extensive dataset enabled the development of a robust graph-based model capable of detailed structural characterization.

The NEGCN framework transformed the initial adjacency graphs, which represented spatial relationships among amino acids, into edge graphs. This transformation allowed us to encapsulate various connectivity types and the spatial arrangement of amino acids,

categorized based on inter-amino acid angles. A four-layer neural network was employed to process these edge graphs and generate comprehensive feature vectors that effectively represented the 3D structure of proteins. Our model utilized three distinct message-passing methods to update node embeddings and produce a detailed and multi-faceted representation of each node. By averaging the cascaded representations across all nodes, we derived a singular, robust representation for the entire protein structure. The NEGCN model was

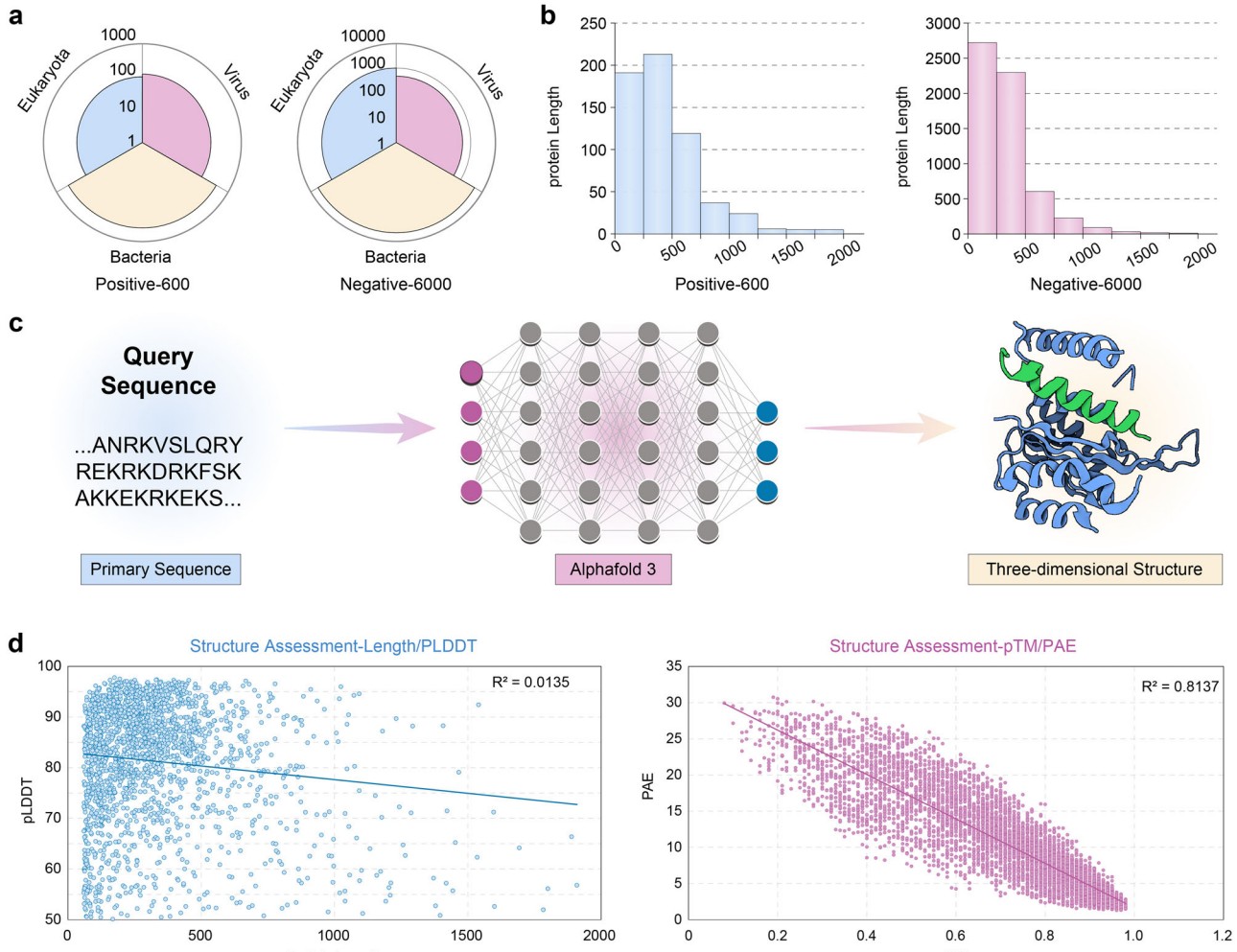

**Fig. 2 | Establishment and evaluation of the protective antigen dataset.**
**a** Pathogen-type distribution of antigens in the positive (Virus: 119, Bacteria: 386, Eukaryota: 95) and negative (Virus: 481, Bacteria: 4493, Eukaryota: 1026) datasets.
**b** Protein length distribution across positive and negative antigens. **c** AlphaFold3-based prediction pipeline for representative antigen structures. Created in BioRender. Zai, X. (2025) https://BioRender.com/g0zw941. **d** Correlation analysis between pLDDT and protein length ($R^2 = 0.0135$). **e** Inverse correlation between PAE and pTM ($R^2 = 0.8137$). Source data are provided as a Source Data file.

trained using a contrastive learning approach to optimize the proximity of positive sample representations to target samples while distancing negative sample representations. This training method employed a contrastive loss function that facilitated the effective differentiation between structurally similar and dissimilar protein representations. The successful pre-training of the NEGCN model established a solid foundation for the downstream machine learning task of classifying and predicting vaccine antigens.

Using the pre-trained NEGCN model, we obtained 960-dimensional feature vectors of the structures for each protein in the antigen dataset. The learned structure feature vectors were then reduced to 27 dimensions via the feature selection pipeline. Subsequently, based on the protein sequence and structure feature vectors in the training set, multiple typical machine learning classification algorithms (i.e., ridge regression, balanced bagging, linear SVC, random forest, and XGBoost) were used for model training[39–44].

The comparative evaluation of machine learning algorithms across viral, bacterial, and eukaryotic pathogens demonstrated the superior performance of XGBoost in antigen prediction tasks (Fig. 3 and Table 1). In bacterial antigen prediction, XGBoost exhibited exceptional discriminative power, attaining the highest Accuracy (0.946) and PR-AUC (0.660), alongside superior F1-score (0.624) and Matthew's correlation coefficient (MCC = 0.598), outperforming

random forest (F1 = 0.524, MCC = 0.507) and linear models (Fig. 3b). For eukaryotic pathogens, XGBoost maintained robust performance with the highest accuracy (0.942) and balanced MCC (0.600), significantly exceeding random forest (MCC = 0.383) in generalization capability (Fig. 3c). For viral targets, XGBoost achieved the highest accuracy (0.867) and Recall (0.458) among all classifiers, coupled with a competitive MCC of 0.532 despite dataset imbalance (Fig. 3d). Across all pathogen categories, XGBoost consistently delivered the strongest integrated metrics, including peak accuracy (0.949), F1-score (0.683), and MCC (0.663), while achieving the highest PR-AUC (0.692), a critical indicator of reliability in imbalanced classification (Fig. 3a). These data collectively validate XGBoost as the optimal model for cross-pathogen antigen prediction, particularly due to its ability to synergistically optimize discriminative power (ROC-AUC) and class-imbalance resilience (PR-AUC) through effective integration of sequence-structural features.

To empirically test our central hypothesis—that there exist learnable, pathogen-agnostic physicochemical and structural features correlated with protective antigenicity that are best captured from a diverse dataset—a rigorous cross-validation analysis was performed. In this analysis, "specialist" models were trained exclusively on pathogen-specific subsets of the data (i.e., a "Virus-only" model, a "Bacteria-only" model, and a "Eukaryota-only" model). The performance of these

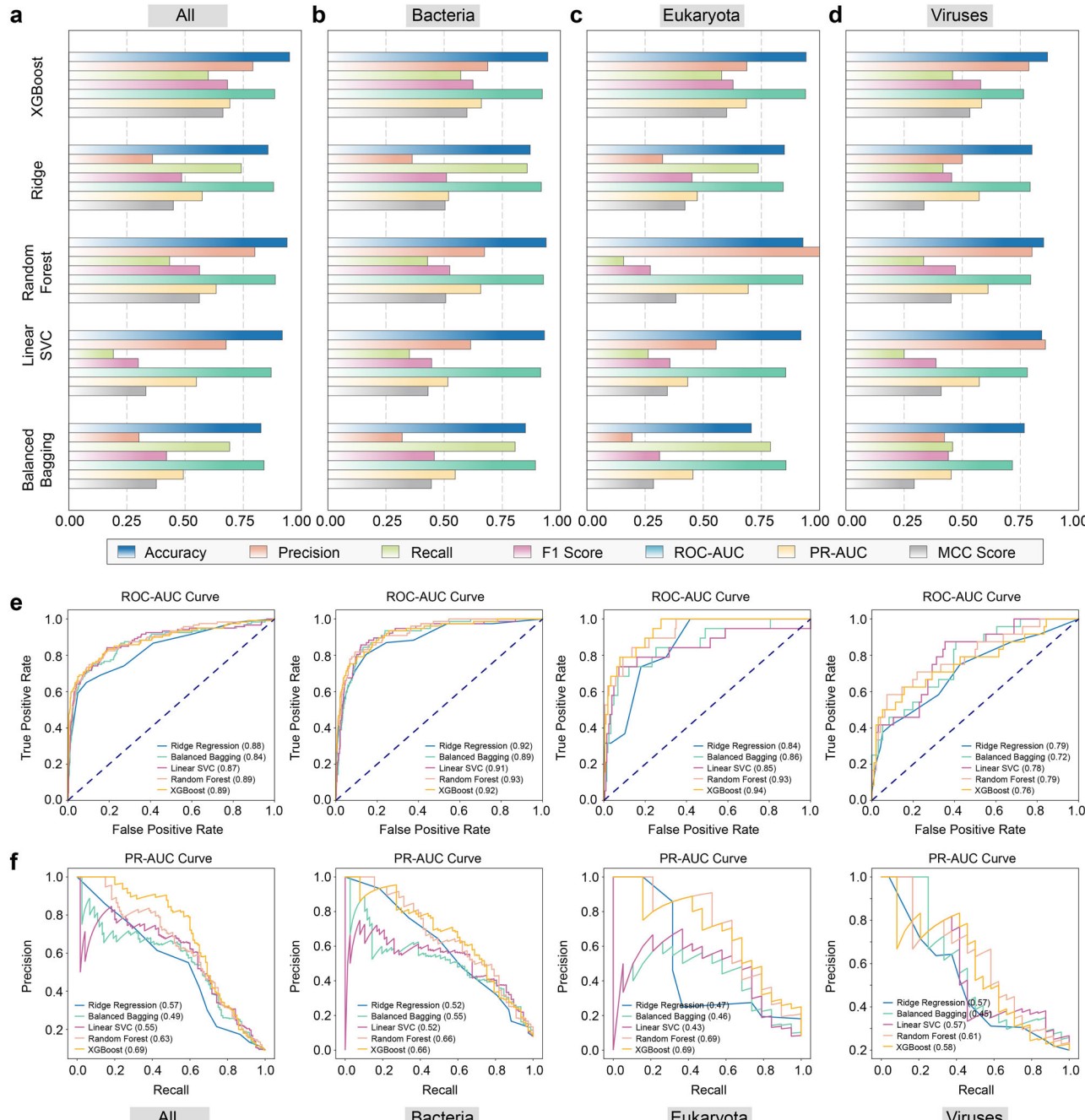

**Fig. 3 | Predictive performance comparison of multiple classification algorithms on the constructed dataset. a–d** Comparative analysis of PLGDL using multiple typical machine learning classification algorithms: i.e., ridge regression, balanced bagging, linear SVC, random forest, and XGBoost. Accuracy, precision, recall, F1 score, ROC-AUC, PR-AUC, and MCC metrics are presented. **e–f**, ROC-AUC and PR-AUC curves are presented. Source data are provided as a Source Data file.

specialist models was then compared against our original "generalist" PLGDL model, which was trained on the complete, combined dataset, when evaluated on each pathogen-specific test set. The results, detailed in Supplementary Table 3, unequivocally demonstrate that the generalist model achieves superior or comparable performance across all pathogen classes, validating our integrated training strategy.

### Evaluation of the PLGDL model on the constructed and third-party datasets

To evaluate the ability of our classification model to effectively distinguish between positive and negative protective antigens in the dataset, we conduct an embedding analysis. Considering that the representation embedding of such pre-trained models cannot

measure the similarity between samples with naive metrics (e.g., Euclidean distance or inner product)[45,46], we used the deep clustering technique for embedding analysis[47]. Specifically, we generate an unsupervised two-dimensional visualization of the embedding space by combining a contrastive learning framework with a neighbor embedding method. A visualization of the results is given in Fig. 4.

All four plots exhibited a clustering phenomenon between positive protective antigen and negative non-protective antigen groups, indicating that the feature space learned by the model contains significant antigen information. The pronounced cluster boundaries in the embedding plots for the antigen database imply a higher degree of separability in the feature space, which directly translates to better classification performance. This suggests that PLGDL can effectively

**Table 1 | Predictive performance comparison of multiple classification algorithms on the constructed dataset**

| Dataset | Methods | Accuracy | Precision | Recall | F1 Score | ROC–AUC | PR–AUC | MCC |
|---|---|---|---|---|---|---|---|---|
| All | Ridge regression | 0.857 | **0.360** | 0.742 | **0.485** | **0.881** | **0.574** | **0.450** |
| | Balanced bagging | 0.886 | 0.419 | 0.650 | 0.510 | 0.840 | 0.493 | 0.463 |
| | Linear SVC | 0.918 | 0.677 | 0.192 | 0.299 | 0.870 | 0.549 | 0.331 |
| | Random forest | 0.939 | **0.800** | 0.433 | 0.562 | **0.888** | 0.633 | 0.561 |
| | XGBoost | **0.949** | 0.791 | 0.600 | **0.683** | 0.885 | **0.692** | **0.663** |
| Bacteria | Ridge regression | 0.870 | 0.363 | **0.857** | 0.510 | 0.917 | 0.519 | 0.504 |
| | Balanced bagging | 0.891 | 0.396 | 0.714 | 0.509 | 0.892 | 0.548 | 0.479 |
| | Linear SVC | 0.931 | 0.614 | 0.351 | 0.446 | 0.915 | 0.516 | 0.431 |
| | Random forest | 0.939 | 0.674 | 0.429 | 0.524 | **0.927** | 0.657 | 0.507 |
| | XGBoost | **0.946** | **0.688** | 0.571 | **0.624** | 0.922 | **0.660** | **0.598** |
| Eukaryota | Ridge regression | 0.849 | 0.326 | **0.737** | 0.452 | 0.844 | 0.475 | 0.422 |
| | Balanced bagging | 0.813 | 0.275 | **0.737** | 0.400 | 0.856 | 0.456 | 0.370 |
| | Linear SVC | 0.920 | 0.556 | 0.263 | 0.357 | 0.855 | 0.433 | 0.346 |
| | Random forest | 0.929 | **1.000** | 0.158 | 0.273 | 0.929 | **0.694** | 0.383 |
| | XGBoost | **0.942** | 0.688 | 0.579 | **0.629** | **0.940** | 0.685 | **0.600** |
| Viruses | Ridge regression | 0.800 | 0.500 | 0.417 | 0.455 | 0.792 | 0.572 | 0.335 |
| | Balanced bagging | 0.808 | 0.526 | 0.417 | 0.465 | 0.716 | 0.453 | 0.354 |
| | Linear SVC | 0.842 | **0.857** | 0.250 | 0.387 | 0.780 | 0.573 | 0.409 |
| | Random forest | 0.850 | 0.800 | 0.333 | 0.471 | **0.794** | **0.611** | 0.452 |
| | XGBoost | **0.867** | 0.786 | **0.458** | **0.579** | 0.764 | 0.583 | **0.532** |

The highest value for each metric is highlighted in bold.

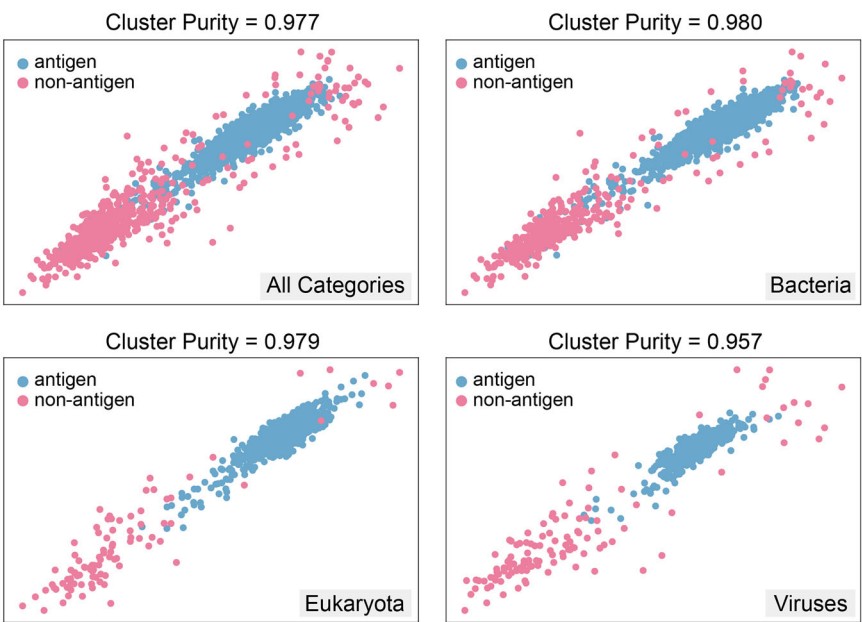

**Fig. 4 | Visualization of the PLGDL embedding on the constructed dataset.** Plots exhibit a clustering phenomenon between positive protective antigen and negative non-protective antigen groups. Source data are provided as a Source Data file.

classify protective antigens and nonprotective antigens based on these features.

Furthermore, we use the metric of cluster purity to quantify the quality of the clusters, i.e., quantifying how homogeneous each cluster is with respect to the ground truth labels. A high cluster purity value means the data points within each cluster predominantly belong to the same ground truth class, implying that the feature space is well-separated. The consistency between the clustering observed in the embedding analysis and the high classification accuracy reported in Table 1 reinforces the validity of the model's approach and its

applicability to real-world pathogen datasets. Notably, the Bacterial category achieved the highest cluster purity (0.980), aligning precisely with its superior classification accuracy in Table 1, a finding that underscores the model's exceptional discriminative capability for bacterial antigen prediction. This robust clustering performance suggests that bacterial antigen features are both distinct and cohesive in the embedding space, enabling highly reliable classification. In contrast, the Viruses category exhibited slightly weaker clustering (purity = 0.957), which is likely attributable to their comparatively limited antigen dataset. The observed cluster homogeneity—particularly

**Table 2 | Predictive performance of our proposed PLGDL model and benchmark methods on a standard dataset**

| Methods | Precision | Recall | F1 score | MCC |
|---|---|---|---|---|
| Vaxign | 0.79 | 0.32 | 0.56 | 0.27 |
| VaxiJen | 0.68 | 0.69 | 0.66 | 0.32 |
| VaxiJen3 | 0.71 | 0.78 | 0.71 | 0.42 |
| Vaxign-ML | 0.75 | 0.81 | 0.76 | 0.51 |
| Vaxi-DL | 0.67 | **0.92** | 0.70 | 0.46 |
| Vaxign-DL | 0.73 | 0.83 | 0.74 | 0.49 |
| PLGDL | **0.80** | 0.84 | **0.82** | **0.60** |

The highest value for each metric is highlighted in bold.

pronounced in bacterial and eukaryotic subgroups—validates the model's ability to encode biologically meaningful protective antigenic signatures.

To further validate our methods, we conducted comparisons using a classical third-party standard dataset (Including 131 positive antigens and 115 negative antigens; Supplementary Data 3) against several benchmark antigens prediction methods: Vaxign, VaxiJen, Vaxijen3, Vaxign-ML, Vaxi-DL and Vaxign-DL[14,15,17,18,48,49]. Vaxign is the most traditional out of the group, and it leverages a rule-based approach combined with bioinformatics tools to predict bacterial vaccine antigens. Unlike Vaxign, VaxiJen does not rely on sequence alignment, which makes it faster and less computationally intensive. Moreover, it focuses on intrinsic properties like hydrophobicity and molecular weight. VaxiJen3 improves upon VaxiJen by incorporating more sophisticated algorithms and larger datasets for better predictive performance. Vaxign-ML represents a hybrid approach that combines the heuristic-based methods of Vaxign with the predictive power of machine learning, thus enabling more accurate antigen prediction. Meanwhile, Vaxi-DL and Vaxign-DL leverage the capabilities of deep learning to capture intricate sequence features that traditional and machine learning models might miss. It is especially effective in dealing with large and complex datasets, and it is a more robust and accurate prediction model.

The results detailed in Table 2 demonstrate that the PLGDL model exhibited superior predictive ability in identifying antigenic candidates for vaccine development; this is evidenced by its Precision, F1 score and MCC. Additionally, the deep learning algorithms based on amino acid sequence features, i.e., Vaxign-DL and Vaxi-DL, also showed promising performance. Overall, these results highlight the robustness and enhanced performance of the PLGDL model, which leverages the complementary strengths of sequence and structural feature fusion.

### Prediction of candidate antigens for Mpox

Mpox virus is an enveloped double-stranded DNA virus with a brick-shaped morphology, and it is one of the largest viral particles[50]. It can propagate within a transmission cycle involving primary hosts, such as squirrels and rats, as well as incidental hosts, including non-human primates (Fig. 5a)[50]. Since 2022, it has continued to spread in Africa and threatens global health security[51]. The recent Mpox outbreak has been declared a "Public Health Emergency of International Concern" by the World Health Organization, marking the second such announcement in two years. Currently available vaccines to combat the outbreak include ACAM2000, JYNNEOS, and LC16m8, all of which are attenuated live vaccinia virus vaccines[52–55]. Limited by safety and accessibility, these vaccines are rarely used in high-risk areas, especially in Africa and other regions[56,57]. There are no approved, safer subunit vaccine options available yet[58]. The Mpox virus has a large genome (>200 K) and expresses a complex array of proteins (>190 ORFs, Supplementary Data 4). Since vaccine antigens for the Mpox virus have not been fully identified, the virus is a typical complex pathogen suitable for real-world validation of prediction models.

The application of the proposed model was rapidly carried out on the Mpox virus. First, the 3D structures of all 190 annotated proteins of the Mpox virus (MPXV-USA-2022-MA001 (ID: ON563414.3)) were predicted, and the output Protein Data Bank (PDB) files were obtained. The pre-trained protein language and geometric deep learning models were used to extract the sequence and structural feature vectors of the 190 proteins. The extracted feature vectors were then input into the trained PLGDL model. Upon running the model, we generated a distribution table of antigen probability scores (Fig. 5b and Supplementary Data 5). Subsequent analysis revealed that fewer than 10% of the proteins achieved a score above 0.5.

A top 10 candidate antigens list was generated and is presented in Table 3. Dominating the ranking, experimentally validated protective antigens—B6R (Rank 1, Score = 0.9194), M1R (Rank 2, Score = 0.8841), A35R (Rank 4, Score = 0.8348), H5R (Rank 6, Score = 0.8247), D8L (Rank 8, Score = 0.7162), and A5L (Rank 9, Score = 0.6887)—are corroborated by their documented roles in viral entry, membrane fusion, and immune modulation[50,59,60]. Among these, three (M1R, B6R, A35R) have already been successfully utilized in the development of mRNA vaccines and achieved preliminary clinical success[61,62]. This validates the efficacy of our antigen prediction model in real-world intricate viral scenarios.

Notably, the model identified potential candidates such as J3R (Rank 3, Score = 0.8836), a chemokine-binding homolog of vaccinia B29R/C23L implicated in immune evasion; A13L (Rank 5, Score = 0.8343), a virion core protease involved in capsid maturation, mirroring vaccinia A12L's proteolytic activity, G10R (Rank 7, Score = 0.7711), a fusion complex component critical for viral membrane integration, and A41L (Rank 10, Score = 0.6782), a Chemokine binding protein reducing infiltration of inflammatory cells into the infected area; both of which lack prior vaccine validation despite their conserved functional roles[63–66].

### Immune evaluation of candidate antigens for Mpox

To verify the accuracy of the model's predictions for Mpox, we successfully expressed four antigens and performed immune evaluation in mouse models, including the validated vaccine antigen controls M1R (Score = 0.8841), two high-scoring candidates (J3R, Score = 0.8836; G10R, Score = 0.7711), and a low-scoring antigen B9R (Score = 0.4434) (Fig. 5c,d)[67]. Immunological results indicated that all candidate antigens could elicit robust specific humoral immunity. Antibody levels reached or surpassed those of the positive control antigen M1R (Fig. 6b). Notably, G10R and B9R demonstrated exceptional performance, with antibody levels reaching 18-fold and 37-fold, respectively, compared to M1R after booster immunization. Subclass analysis of the antibodies revealed a balanced Th1 and Th2 immune response, thus indicating the broad immunogenic potential of these antigens.

However, critical divergence emerged in protective efficacy: G10R generated neutralizing antibody levels comparable to M1R in microneutralization assays against ectromelia virus (ECTV), a surrogate for Mpox, and conferred partial protection in lethal challenge models (15-day survival vs. 7-day mortality in controls)[50]. Crucially, sequence alignment confirmed exceptionally high conservation (>96% identity) between the potential MPXV antigen (G10R) and its ECTV homolog (Supplementary Fig. 3), strongly supporting the biological relevance of the observed immune responses. In stark contrast, B9R, despite its exceptional immunogenicity as a secretory protein, failed to elicit protective effects, validating the model's capacity to discriminate between immunogenic yet non-protective antigens and true protective candidates[67].

## Discussion

The field of vaccine development would greatly benefit from artificial intelligence models that predict antigens[7,8,16]. Traditionally, these models have relied on manually curated features derived from protein

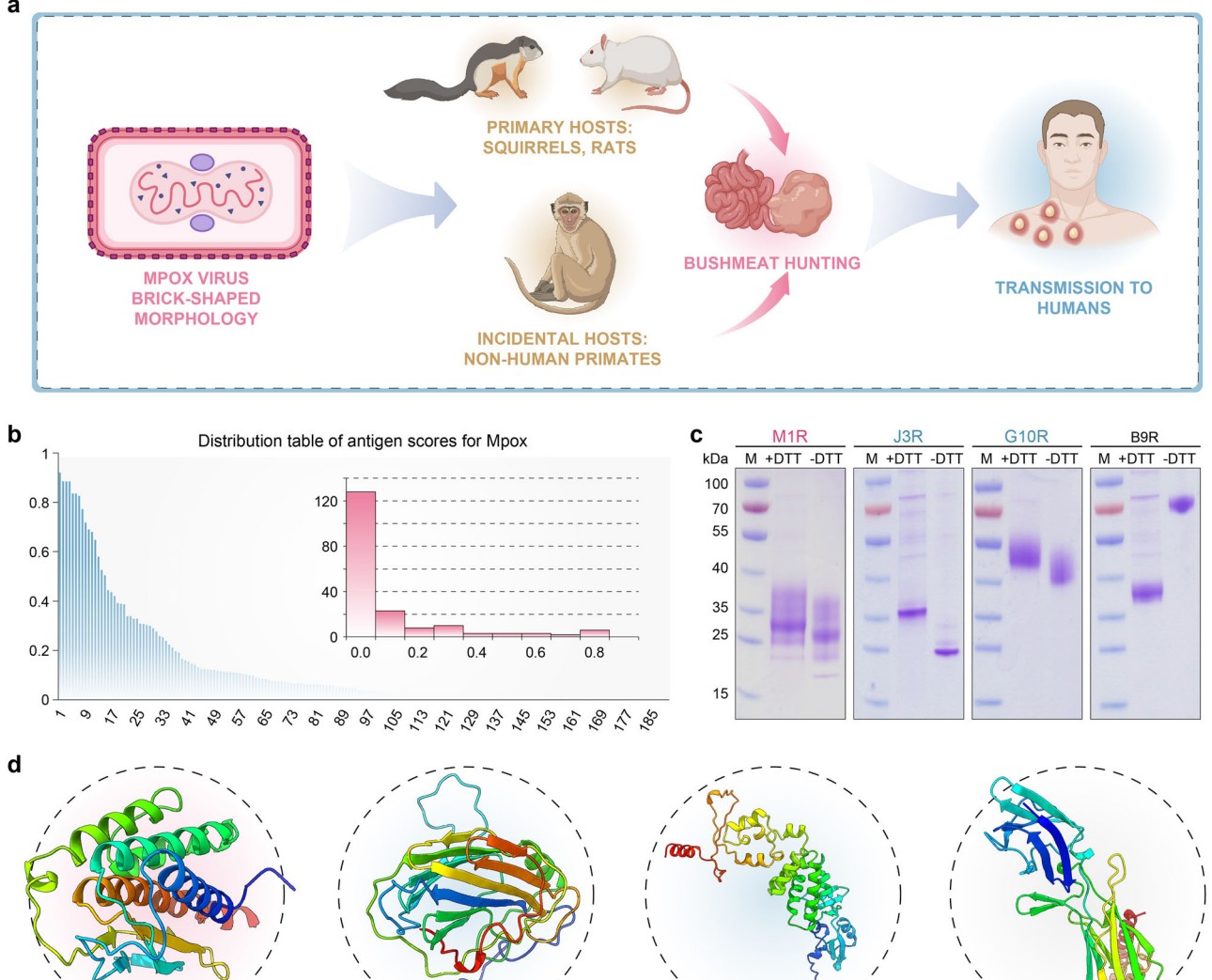

**Fig. 5 | Prediction of novel candidate antigens for Mpox. a** The complicated transmission cycle of the Mpox virus. Created in BioRender. Zai, X. (2025) https://BioRender.com/jushumr. **b** Distribution table of antigen scores for Mpox. **c** Expression and SDS-PAGE identification of candidate antigens. The experiment was repeated independently three times with similar results. **d** Protein structure modeling of candidate antigens, with color-coded annotations: brown = validated vaccine antigen controls, green = model-prioritized high-scoring candidates, black = low-scoring negative controls. Source data are provided as a Source Data file.

amino acid sequences, which has limited their ability to uncover more complex and deeper biomolecular correlations associated with antigenicity[12,13]. Recent advancements in protein language models, such as ESM[23], have shown promise in effectively representing and extracting feature information embedded within one-dimensional amino acid sequences[24]. These models input protein sequences and learn the underlying biochemical properties, secondary and tertiary structures, and functional rules within the sequences, which may aid in the development of RV-based antigen prediction models.

Moreover, we propose that similarity in protein structure, rather than sequence, may better characterize the inherent properties of antigens. The structure of a protein provides much more information than the one-dimensional amino acid sequence, as it ultimately determines the biological characteristics and physicochemical parameters of the protein. Establishing a machine learning classification prediction model for antigens based on protein structural features is expected to break through the limitations of existing models based solely on one-dimensional amino acid sequences.

The 3D structures of antigens can be predicted using cutting-edge intelligent algorithms, such as AlphaFold2 and AlphaFold3 [27,28]. To better represent the 3D structures of proteins, we used an improved geometric deep learning model called NEGCN. This model incorporates both local and global structural information by passing messages between neighboring nodes in a protein graph[32]. The NEGCN model generalizes the traditional CNN approach by incorporating structural information encoded in the protein graph, thereby providing a more accurate representation of protein structures.

In this work, we finally developed a framework termed PLGDL, which integrates features extracted from both a protein language model and a graph neural network. The protein language model was employed to capture sequential information from the protein sequences, while the geometric deep learning model NEGCN was designed to effectively represent the 3D structural features of proteins.

By integrating 1D and 3D protein features, PLGDL demonstrated a significant improvement compared to existing antigen prediction

**Table 3 | Top 10 candidate antigens for Mpox output by the PLGDL model**

| Rank | Protein ID | Protein name | Score | Known antigens |
|------|-----------|--------------|-------|----------------|
| 1 | URK20605.1 | B6R EEV type-1 membrane glycoprotein, protective antigen (Cop-B5R), complement control protein-like palmitated 42 kDa glycoprotein located both on the membranes of infected cells and on EEV envelop, similar to Vaccinia virus strain Copenhagen B5R | 0.9194 | Yes |
| 2 | URK20517.1 | IMV membrane protein (Cop-L1R) M1R myristylated IMV surface membrane protein similar to Vaccinia virus strain Copenhagen L1R | 0.8841 | Yes |
| 3 | URK20629.1 | CC-chemokine binding Chemokine binding protein (Cop-C23L) J3R similar to Vaccinia virus strain Copenhagen B29R | 0.8836 | No |
| 4 | URK20584.1 | A35R EEV envelope glycoprotein, needed for the formation of actin-containing microvilli and cell-to-cell spread of virion EEV membrane phosphoglycoprotein, C-type lectin-like domain (Cop-A33R) interacts with VAC A36R, similar to Vaccinia virus strain Copenhagen A33R | 0.8348 | Yes |
| 5 | URK20560.1 | A13L Virion core and cleavage processing protein (Cop-A12L) similar to Vaccinia virus strain Copenhagen A12L virion core protein | 0.8343 | No |
| 6 | URK20532.1 | Ca2 + -binding motif H5R VLTF-4 (late transcription factor 4) (Cop-H5R) similar to Vaccinia virus strain Copenhagen H5R virosome-associated late gene transcription factor, VLTF-4 | 0.8247 | Yes |
| 7 | URK20516.1 | Entry/fusion complex component, myristylprotein (Cop-G9R) G10R myristylated protein similar to Vaccinia virus strain Copenhagen G9R | 0.7711 | No |
| 8 | URK20450.1 | ANK-containing protein D8L | 0.7162 | Yes |
| 9 | URK20552.1 | 39 kDa immunodominant virion core protein needed for the progression of IV to infectious IMV. 39 kDa virion core protein (Cop-A4L) A5L similar to Vaccinia virus strain Copenhagen A4L | 0.6887 | Yes |
| 10 | URK20590.1 | A41L Chemokine binding protein (Cop-A41L) secreted protein, reducing infiltration of inflammatory cells into the infected area, similar to Vaccinia virus strain Copenhagen A41L | 0.6782 | No |

methods. Multiple classification algorithms were employed for model training. Evaluation metrics indicated that XGBoost exhibited the best prediction performances, and the embedding analysis results indicate that the feature space learned by the model contains significant antigen information. The pronounced cluster boundaries in the embedding plots for the antigen dataset suggest a higher degree of separability in the feature space, which directly translates to better classification performance. We can thus conclude that this model is applicable to different types of pathogens and that it can learn the characteristics of known protective antigens to predict potential vaccine antigens. Further comparisons against benchmark methods (Vaxign, VaxiJen, VaxiJen3, Vaxign-ML, Vaxi-DL, and Vaxign-DL) using a standard third-party dataset further validated our model[15]. The PLGDL model demonstrated superior predictive ability with the highest precision, weighted F1 score, and MCC. These results highlight the robustness and enhanced performance of the PLGDL framework because it leverages the complementary strengths of sequence and structural feature fusion.

A deliberate and core methodological choice in this study was the use of a unified training dataset comprising antigens from viral, bacterial, and eukaryotic pathogens. This approach is grounded in the principles of structural vaccinology, which posits that the outcome of protection is often correlated with intrinsic, learnable physicochemical and structural properties of the stimulus—the protein antigen itself—irrespective of the specific downstream immune pathway it triggers. These structural commonalities suggest the existence of a learnable, pathogen-agnostic structural feature correlated with protective antigenicity. The PLGDL framework, by synergistically combining a protein language model to capture sequence-level context and a geometric deep learning model to decipher 3D structural information, is explicitly designed to learn these features.

To assess practical, real-world applications, we utilized the PLGDL model to predict candidate antigens for the ongoing outbreak pathogen: Mpox virus. There is an urgent need for the identification of novel candidate antigens to support the development of Mpox vaccines. The top 10 predicted antigens not only included six previously reported vaccine targets but also identified several candidate antigens such as J3R and G10R. Subsequent immunological evaluations were carried out in mouse models to assess their potential. The results confirmed that these antigens could elicit higher levels of binding antibodies compared to the positive control antigen M1R. Notably,

G10R also demonstrated significant neutralizing effects. Most importantly, through using a lethal orthopoxvirus challenge model, we found that G10R conferred partial protection. Our follow-up analysis of the lead candidate G10R revealed critical functional validation of its protective potential. Isolation of G10R-specific monoclonal antibodies (mAbs) via hybridoma technology identified a high-affinity neutralizing antibody (Supplementary Fig. 4). This neutralization capacity, aligning with prior evidence linking in vitro neutralization to in vivo protection in orthopoxvirus models, establishes G10R as a new, definitively confirmed protective antigen within its class, distinguishing its specific immunogenicity from nonspecific immune artifacts.

The G10R protein of the Mpox virus is a myristoylated protein similar to the vaccinia virus strain Copenhagen G9R. Previous studies showed that G9R is one of eight proteins associated in a putative entry-fusion complex of the vaccinia virus[65]. G10R's potential as a vaccine target is underscored by its immunogenicity and ability to confer partial protection in mouse models. Its significant neutralizing effects and low sequence homology to human proteins make it a promising component for a subunit vaccine (Supplementary Data 6). Considering the efficacy of M1R and other classic antigens, there is a strong rationale for exploring the combined use of G10R with these antigens. Such a combination could potentially enhance the overall protective efficacy of the Mpox vaccine by targeting multiple stages of the viral lifecycle, thus providing broader immunity. These findings underscore the efficacy of the PLGDL model employed in this study for the rapid identification of potential vaccine antigens for complex pathogens. Further evaluation of more candidates in advanced studies is warranted to assess their suitability for inclusion in future vaccine formulations.

In this study, we have deliberately focused on predicting protective antigens rather than merely immunogenic ones, as suggested, recognizing the critical distinction between these concepts. While immunogenicity refers to an antigen's ability to induce detectable adaptive immune responses (either humoral or cellular), protection specifically denotes the capacity to confer resistance against pathogenic challenge when used as an immunogen, as demonstrated through rigorous in vivo animal protection studies[2,34]. This distinction is crucial because immunogenicity represents a necessary but insufficient condition for protection, as evidenced by numerous vaccine candidates that show strong immunogenicity but fail to provide actual protection in clinical trials.

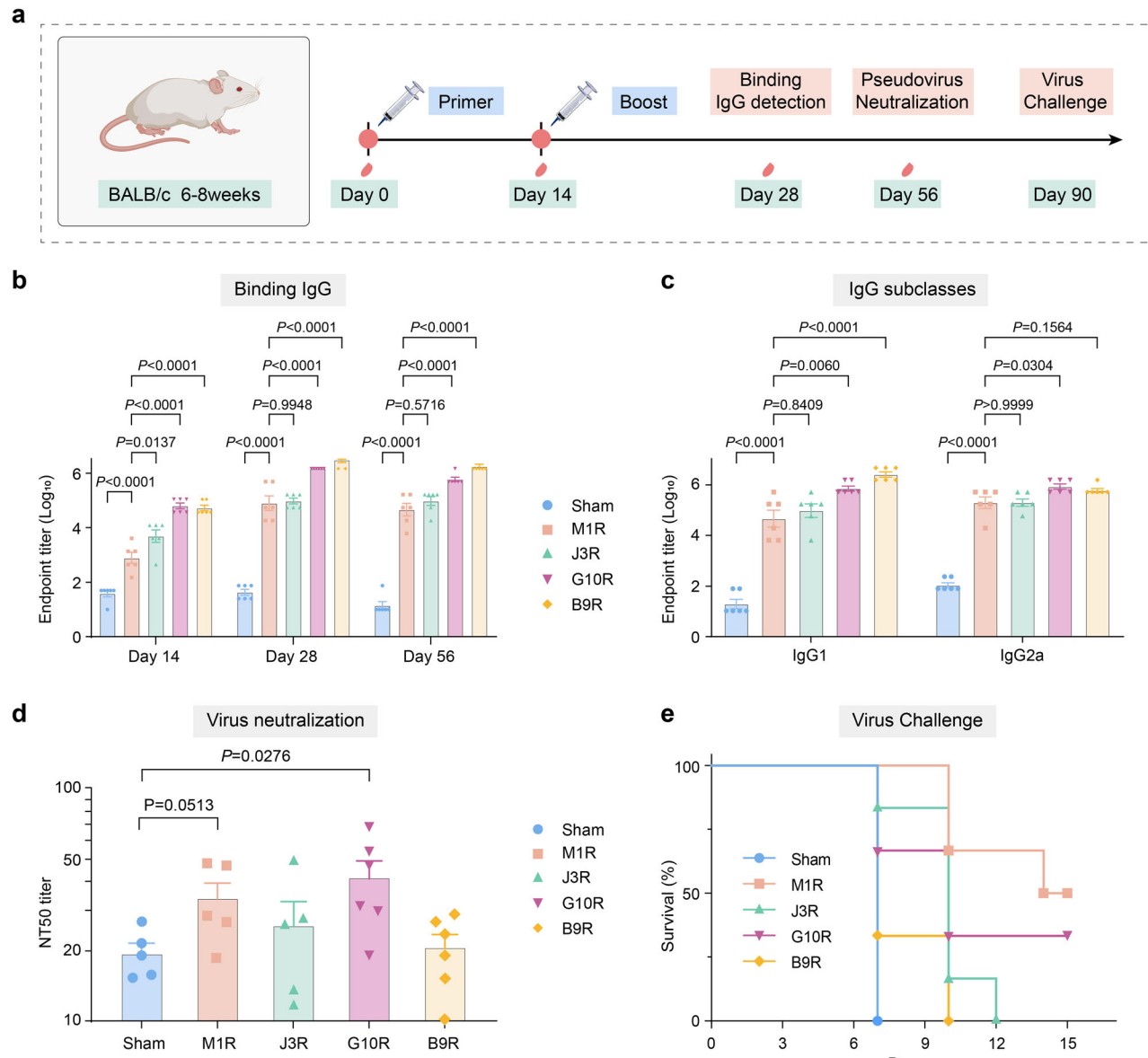

**Fig. 6 | Immune evaluation of candidate antigens for Mpox. a** Schematic of the immunization assessment process. Specific pathogen-free (SPF) female BALB/c mice (6–8 weeks old, $n = 6$ biologically independent mice per group) were immunized intramuscularly with 100 μL of vaccine formulations at day 0 and day 14. Created in BioRender. Zai, X. (2025) https://BioRender.com/jrjco9u. **b** Specific total IgG antibody levels in mouse immune serum (mean ± SEM, $n = 6$ biologically independent mice per group). **c** Specific IgG antibody subclasses in mouse immune serum (mean ± SEM, $n = 6$ biologically independent mice per group). **d** Detection of neutralizing antibodies against ectromelia virus (mean ± SEM, $n = 6$ biologically independent mice per group). **e** Protective efficacy in a lethal orthopoxvirus challenge. One-way analysis of variance (ANOVA) followed by Tukey's post hoc test (for multiple group comparisons) and unpaired two-tailed t-test (for pairwise comparisons) were used to analyze differences between groups. Exact p-values for all comparisons are provided in the figure.

Our research hypothesis stems from the intriguing observation that many successful vaccine target antigens, despite substantial sequence divergence, share common structural features. For instance, viral envelope glycoproteins (e.g., influenza HA, HIV Env, RSV F protein) frequently adopt trimeric conformations with distinct head and stalk domains, while Gram-negative bacterial outer membrane proteins often exhibit conserved beta-barrel structures[68,69]. These structural commonalities suggest that protective antigens may share conserved structural motifs that are not apparent at the sequence level. Our study specifically investigates whether such structural features can be captured by advanced protein structure prediction tools (e.g., AlphaFold) and effectively learned by graph neural networks (GNNs) to predict vaccine target potential.

The methodological innovation of our approach lies in leveraging sequence and structural insights to develop a computational framework. By combining state-of-the-art protein structure prediction with graph-based deep learning techniques, we aim to identify and extract structural patterns that correlate with protective antigenicity across diverse pathogens. This structure-centric approach may help to address several limitations of sequence-only based methods, particularly in handling domain-specific variations in immunogenicity and protection.

We also identified several potential challenges and limitations of the present study. Firstly, the relatively small number of well-characterized protective antigens in the positive database may constrain the complexity and generalizability of models trained on such datasets. Additionally, existing protective antigen databases may contain biases, as they reflect protection observed under specific experimental conditions (particularly animal models, pathogen

strains, immunization protocols)[34]. Antigens effective in one model system may not demonstrate equivalent protection in others or in humans[70]. Differences in genetics, physiology, and pathogen tropism mean that no single animal model can perfectly recapitulate human immune responses. Furthermore, such databases might overlook antigens whose protective mechanisms are difficult to assess using standard animal models, or those effective in humans but not in conventional test models. These considerations highlight the ongoing challenges in developing truly predictive models of protective immunity. Given the intricate and dynamic nature of the immune system, forecasting the antigenicity of a vaccine or antigen is a formidable task. The PLGDL model is not intended to be a definitive predictor of clinical success in humans. Rather, its purpose is to serve as a powerful, high-throughput in silico screening and prioritization tool within the modern RV pipeline. The core value of such a tool lies in its ability to analyze an entire pathogen proteome—comprising hundreds or thousands of proteins—and identify a small, manageable list of the most promising candidates for expensive and labor-intensive experimental validation. As demonstrated by our Mpox case study, this dramatically accelerates the discovery phase.

Moreover, the PLGDL model may necessitate substantial computational resources when incorporating large-scale protein structural data or utilizing advanced machine learning techniques. Consequently, this could restrict the model's applicability in settings where computational resources are limited. In addition, we reduced the information in 3D space to a 2D plane through contact graphs, which inevitably leads to a loss of some structural information. How to make better use of the advantages of automatic feature extraction in deep learning models to extract richer and more important structural features for antigen prediction is a focus of our future work.

In summary, RV is the critical first step in modern rational vaccine design, with its core value lying in efficiently navigating vast biological search spaces to identify promising candidates for experimental validation. As exemplified by the present study, which introduces a previously uncharacterized protective antigen prediction framework, PLGDL, that leverages the synergistic use of protein language and geometric deep learning models. This integrated approach not only offers substantive insights but also represents a significant advancement in vaccine development, as demonstrated by its successful application to the ongoing Mpox virus outbreak, thereby reinforcing the role of RV in accelerating the discovery of protective targets and shaping the future of precision vaccinology.

## Methods

### Antigen dataset establishment

**Positive dataset construction.** The positive dataset was rigorously curated primarily from the Protegen Database, which provides experimentally validated protective antigens meeting one of two criteria: (1) demonstrated in vivo protective efficacy (e.g., statistically significant survival improvement in pathogen challenge models), or (2) induction of antigen-specific immune responses directly associated with protective outcomes (e.g., neutralizing antibodies with confirmed pathogen inhibition)[34]. Additional protective antigens were identified through PubMed-curated studies that met stringent inclusion criteria: demonstrated in vivo protection. The CD-HIT program (V4.8.1), which clusters proteins at various sequence identity levels, was utilized to eliminate redundant sequences with amino acid homology exceeding 50%[71]. This process yielded a dataset of 600 unique vaccine antigens comprising 119 viral, 386 bacterial, and 95 eukaryotic antigens.

**Negative dataset construction.** The negative dataset was constructed by extracting all annotated protein sequences from UniProt-listed pathogens corresponding to those represented in the positive antigen dataset. To ensure rigorous quality control, sequences underwent a two-step homology reduction process. First, intra-dataset redundancy

was minimized by applying a 50% sequence identity cutoff using CD-HIT (V4.8.1), consistent with the homology threshold applied to the positive dataset. Subsequently, all sequences exhibiting >20% sequence similarity to any positive antigen were excluded through BLASTP (V2.14.0) alignment ($E < 1e-5$), effectively eliminating potential cross-dataset biases. This dual-filter approach yielded 6,000 non-redundant negative samples (composition: Viral 481, Bacterial 4493, Eukaryotic 1026; Fig. 2a), intentionally maintaining a 1:10 positive-to-negative ratio to reflect the natural scarcity of protective antigens.

**Quality assessment of the structures.** The structural prediction of the antigens was performed using AlphaFold3 (release 3.0.1), a state-of-the-art deep learning algorithm for protein structure prediction[28]. The input sequences for the predictions were derived from both the positive and negative datasets. Each antigen sequence was processed individually to generate its three-dimensional structure. Once the structural predictions were generated, the corresponding PDB files for each protein were obtained. These files contain the atomic coordinates of the proteins' structures, which are essential for subsequent quality assessments. Structural confidence was evaluated using pLDDT (Predicted Local Distance Difference Test), pTM (Predicted TM Score), and PAE[28].

**Feature extraction based on protein language models for characterizing protein sequences.** The evaluation of three advanced protein language models for sequence feature extraction was conducted: ESM-2, ProTrans, and AMPLIFY[23,36,37]. ESM-2, pre-trained via masked language modeling on the UniRef50 database, captures intricate sequence patterns and relationships, generating 1280-dimensional contextual embeddings per protein sequence. ProTrans produces 1024-dimensional context-aware embeddings that simultaneously encode local residue interactions and global sequence properties, demonstrating strong performance in protein function prediction and structural inference tasks. AMPLIFY employs a hybrid architecture integrating GNNs and attention mechanisms to model sequence-structure relationships, yielding 960-dimensional embeddings optimized for higher-order feature extraction.

### Establishing the neighborhood enhanced graph convolutional network (NEGCN) for characterizing protein structures

**Data acquisition and pre-processing.** High-quality protein structure data totaling 805,000 entries were obtained from the AlphaFold Protein Structure Database[38]. This comprehensive dataset provides a solid foundation for developing robust graph-based models to characterize protein structures comprehensively.

**Graph construction.** We implemented a graph construction approach to represent protein structures spatially. Initially, an amino acid from a given protein (protein A) was selected randomly to serve as the center of a radius (R) (Fig. 1). All amino acids within this radius were included to form an adjacency graph that was then designated as the target sample. Subsequently, another amino acid from protein A was similarly used to generate a corresponding positive sample graph. Conversely, an amino acid from a different protein (protein B) served as the center for constructing a negative sample graph. In these graphs, nodes represent individual amino acids and edges denote spatial adjacency between pairs of amino acids.

**Graph editing via random masking.** To introduce variability and test the robustness of our model, we applied a random mask method whereby a predetermined proportion of edges was randomly omitted from the target, positive, and negative sample graphs. This resulted in modified graph structures.

**Graph transformation and feature extraction.** Our NEGCN model transforms the initial adjacency graph (graph A) into an edge graph

(graph B). This transformation encapsulates connectivity through different adjacency relationships as edges, cascades node attributes, and categorizes edge types based on the angles between amino their spatial positions. This enables precise classification of edge types into various categories according to their spatial positions.

**Neural network architecture.** A four-layer neural network was constructed to process the edge graph by cascading outputs from each layer. In this way, we can generate a comprehensive feature vector representing the 3D structure of the protein.

**Node embedding updates.** Three distinct message-passing methods were utilized to update each node in the edge graph, as shown in Eq. 1.

$$
\begin{aligned}
h_i^m &= \sum_{\theta \in T} W_\theta \sum_{j \in N_i^\theta} \left( h_{i,1}^m + h_{i,2}^m + h_{i,3}^m \right) \\
h_{i,1}^m &= \sum_{\tau \in \hat{k}} W_\tau \sum_{j \in N_i^\tau} h_j^{m-1} \\
h_{i,2}^m &= W_k \sum_{j \in N_i^k} h_j^{m-1} \\
h_{i,3}^m &= W_r \sum_{j \in N_i^r} h_j^{m-1}
\end{aligned}
\tag{1}
$$

Here, $m$ denotes the network layer, and $N_j^\theta$ denotes the neighbors of node $i$ in the $\tau$th quadrant among eight quadrants $T$. $h_{i,1}^m$ updates node $i$'s embedding according to its neighbors among $\hat{k}$ hops, $h_{i,2}^m$ updates node $i$'s embedding according its $k$ nearest neighbors, and $h_{i,2}^m$ updates node $i$'s embedding according its neighbors within a distance of $r$. The embeddings from were then cascaded to form a detailed representation for each node. Subsequently, representations from different layers were cascaded and averaged across all nodes $N$ to derive a singular representation for the entire graph (G): $G = avg\left( \sum_{i=1}^{N} ||_{n=1}^{m} h_i^n \right)$

**Model training and contrastive learning.** The NEGCN model was trained using a contrastive loss function, which facilitates learning by optimizing the representations of positive samples to be closer to the target samples and distancing those of the negative samples in a high-dimensional space.

$$
L = -\log \frac{\exp(sim(G_t, G_P)/\tau)}{\sum_{i=1}^{B} \exp(sim(G_t, G_i)/\tau)}
\tag{2}
$$

where $G_t$, $G_P$ denotes the embeddings of the target node and the positive sample for the target node, $B$ denotes the batch size, and $sim$ computes the cosine similarity. This approach ensures effective pre-training of the model and sets a foundation for subsequent applications in machine learning classification and prediction tasks. Following GearNet[30], we trained our NEGCN for 50 epochs to guarantee a good result. The learning rate was $2 \times 10^{-4}$.

**Machine learning framework that simultaneously leverages protein language and geometric deep learning models**
**Feature selection and model training.** Our PLGDL framework integrates ESM-2 and pre-trained NEGCN to derive complementary representations: 1280-dimensional sequence embeddings and 960-dimensional structural embeddings. To optimize the feature set, we first applied principal component analysis coupled with empirical performance validation (Supplementary Fig. 2) to determine the optimal dimensionality thresholds. For sequence embeddings, the 255-dimensional configuration achieved best performance in F1 score and MMC, while structural embeddings exhibited optimal discriminative power at 27 dimensions, confirming the balance between information retention and redundancy minimization. Based on the refined 255-

dimensional sequence and 27-dimensional structural embeddings, we trained the PLGDL model using five classification algorithms (Ridge Regression, Balanced Bagging, Linear SVC, Random Forest, and XGBoost), and XGBoost achieved the best performance. This workflow ensures maximal leverage of both sequential patterns and structural determinants for antigen prediction.

**Model evaluation.** The efficacy of each model was assessed using a test set and accuracy, precision, recall, F1 score, ROC-AUC, PR-AUC, and Matthew's correlation coefficient (MCC) evaluation metrics. The metrics are defined below.

- **Accuracy:** $\frac{TP + TN}{TP + TN + FP + FN}$
- **Precision:** $\frac{TP}{TP + FP}$
- **Recall:** $\frac{TP}{TP + FN}$
- **F1 Score:** $2 \times \frac{\text{Precision} \times \text{Recall}}{\text{Precision} + \text{Recall}}$
- **ROC–AUC:** Area under the receiver operating characteristic curve
- **PR–AUC:** Area under the precision–recall curve
- **MCC:** $\frac{(TP \times TN) - (FP \times FN)}{\sqrt{(TP + FP) \times (TP + FN) \times (TN + FP) \times (TN + FN)}}$

TP, FP, TN, and FN are the number of true positive samples, false positive samples, true negative samples, and false negative samples, respectively. These metrics were chosen to provide a comprehensive overview of the models' predictive performances by capturing both the effectiveness and reliability of predictions. Using a third-party standard dataset, additional comparisons are made with several benchmark antigen prediction models: Vaxign, VaxiJen, VaxiJen3, Vaxign-ML, Vaxi-DL, and Vaxign-DL.

**Cross-validation on pathogen-specific subsets.** To empirically evaluate the hypothesis that a model trained on a diverse, combined dataset would achieve superior or comparable performance to models trained on homogeneous, pathogen-specific datasets, a cross-validation analysis was conducted. The full dataset, comprising 600 positive and 6000 negative samples, was partitioned while maintaining the original training-test split ratio. The training data was further subdivided into three mutually exclusive sets based on pathogen origin: "Virus-only" "Bacteria-only" and "Eukaryota-only". Three separate "specialist" XGBoost models were trained, one on each of these pathogen-specific training sets. For evaluation, each specialist model was assessed only on the test set corresponding to its pathogen type (e.g., the Virus-only model was tested on the Virus test set). The original generalist model, trained on the combined "Virus+Bacteria +Eukaryota" training set, was evaluated on all three pathogen-specific test sets to provide a direct performance comparison across the same seven metrics: Accuracy, Precision, Recall, F1 Score, ROC-AUC, PR-AUC, and MCC.

**Embedding analysis.** In this study, we used a deep clustering technique for embedding analysis[47]. Initially, embeddings were extracted from the pre-trained model for all samples. These embeddings were then refined using contrastive learning, which enhances the separability of similar and dissimilar samples in the latent space. The contrastive learning framework optimizes the embedding space by bringing similar samples closer while pushing dissimilar samples apart through a contrastive loss function. Subsequently, a neighbor embedding method was used, specifically *t*-distributed stochastic neighbor embedding or Uniform Manifold Approximation and Projection, to reduce the high-dimensional embeddings to a two-dimensional space. This step facilitates unsupervised visualization, allowing us to observe the clustering patterns in the embedding space. The resulting 2D visualization effectively highlights the clustering phenomenon, providing insights into the model's ability to distinguish between different sample categories. Subsequently, we use the metric of cluster purity to quantify the quality of the clusters, i.e., quantifying

how homogeneous each cluster is with respect to the ground truth labels.

## Prediction of candidate antigens for Mpox

**Antigen prediction.** The structures of all 190 proteins of the Mpox virus epidemic strain (MPXV USA 2022 MA001) are predicted, and the output PDB files are obtained. The sequence and structural feature vectors are extracted using the pretrained protein language and geometric deep learning models, and then input into the previously established PLGDL model. Note that the classifier model was not trained with known Mpox antigens to avoid overfitting. The output is a list of the top candidate vaccine antigens for Mpox.

**Antigen expression.** The selected candidate antigens of Mpox were prepared using the eukaryotic expression system. Briefly, the amino acid sequence of the target antigens was selected, and then the tissue plasminogen activator signal peptide and Hexa-His tag were added to its N-terminus. Frequently used mammalian cell codons were used to optimize expression. The gene sequence was inserted into the pcDNA3.1 eukaryotic expression vector (Invitrogen, Cat V79520, USA). We used the Expi293F mammalian cell expression system to express the protein. Next, 72 h after transfection, the cell culture medium was centrifuged ($3000 \times g$, 15 min), and the retained supernatant was filtered with a 0.45 μm syringe filter to remove cell debris. The His-tagged recombinant mature proteins were then purified using a 5-ml HisTrap HP (GE Healthcare, Cat 17524801, Sweden) chromatography column according to the manufacturer's instructions. The purity of the protein antigens was further evaluated using SDS-PAGE.

**Immunity evaluation of candidate vaccine antigens for Mpox immunization assessment process.** The experiments involving animals were approved and carried out according to the Institutional Animal Care and Use Committee guidelines of the Laboratory Animal Center (IACUC-DWZX-2024-P008). Specific pathogen-free female BALB/c mice (6–8 weeks) were purchased from Charles River and were used for intramuscular immunization. Mice were housed in individually ventilated cages. Housing conditions were maintained at a controlled temperature ($22 \pm 2\,°C$), humidity ($55 \pm 10\%$), and a 12-h light/dark cycle, with ad libitum access to food and water. Each mouse was injected with 100 μL of vaccine sample, including 5 μg protein antigen, 50 μg alum adjuvant (AH, Brenntag Biosector, Denmark) with 25 μg CpG (Takara, Japan), or PB solution (Sham group). The mice were immunized at day 0 and day 14 and bled from the tail vein at day 0, day 14, day 28, and day 56. The blood was centrifuged at room temperature to obtain serum for subsequent immunological evaluation.

**Binding IgG antibody.** Mouse serum antibody (IgG) levels were detected by enzyme-linked immunosorbent assay. First, the antigen protein was diluted to 2 μg/ml in a carbonate solution. Then, the 96-well plate (Solarbio, China) was coated with 100 μL per well and incubated overnight at 4 °C. PBST was used to wash the 96-well plate three times; the liquid inside the plate was discarded, 100 μL blocking solution (2% BSA dissolved in PBST) was added per well, and the plate was incubated at 37 °C for 1 h. The cleaning step was repeated, and a 100 μL dilution (0.2% BSA dissolved in PBST) with gradient diluted serum was added to the plate, which was incubated at 37 °C for 1 h and washed three times. Next, HRP-conjugated goat anti-mouse IgG (Abcam, Cat ab97265, UK, 1:10,000 dilution) was added to the plates, which were incubated at 37 °C for 1 h and washed with PBST three times. Then, 100 μL of TMB substrate solution (Solarbio, China) was added to each well, and the plates were incubated in the dark for 6 min. Finally, 50 μL of stop solution (Solarbio, China) was added to end the color reaction. Measurements were made at 450 nm/630 nm (SPECTRA MAX 190, Molecular Device, USA). The endpoint titers were calculated as the dilution that exceeded the 2.1-fold value of the background.

**Ectromelia virus neutralization.** The assessment of serum neutralizing activity against live ECTV was conducted through a micro-neutralization assay[72]. Serial dilutions of serum were incubated with tissue culture infective dose 50% of ECTV at 37 °C for 1 h. Following incubation, the serum-virus complexes were introduced to pre-plated BS-C-1 cells (ATCC, Cat CCL-26) monolayers cultivated in 96-well plates and subjected to a further 48 h of incubation. After that, cells treated with ECTV were stained with 0.05% crystal violet for 40 min. Optical density was measured at 570 nm/630 nm after adding the decolorization solution.

**Ectromelia virus challenge.** On day 90, mice were challenged with a lethal dose of ECTV[72]. The viral stock was propagated and titrated on BS-C-1 cells using plaque assays to ensure accurate quantification of the viral load. Each mouse received an intranasal inoculation with a 25 μL suspension containing 200 plaque-forming units of ECTV. Following inoculation, the mice were monitored daily for a period of 15 days for clinical signs of infection. The primary outcome measure was the percentage of survival, calculated at the end of the observation period. This survival data was used to evaluate the protective efficacy of the vaccine antigens against the lethal viral challenge.

**Statistical analysis.** Data analysis was performed using Excel and GraphPad Prism 9.0. Antibody titer data were log10-transformed and analyzed as independent measurements from distinct biological replicates ($n = 6$ mice per group). One-way analysis of variance followed by Tukey's post hoc test (for multiple group comparisons) and unpaired two-tailed t-test (for pairwise comparisons) were used to analyze differences between groups. Data are presented as mean ± SEM (error bars in figures).

## Reporting summary

Further information on research design is available in the Nature Portfolio Reporting Summary linked to this article.

## Data availability

All data supporting the findings of this study, including raw datasets, processed results, and source data, are available within the main text, Supplementary Information, or Supplementary Data files or from the corresponding author on request. Source data are provided with this paper.

## Code availability

The code and data required to reproduce the results in this study are openly available via the repository: https://github.com/yunxiangz/PLGDL. This repository includes the scripts for the final XGBoost classifier and the complete, documented source code for the NEGCN model used to generate the protein structural embeddings.

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

## Acknowledgements

We thank Prof. Lihua Hou for proofreading and editing the manuscript. This work was partially funded by the National Natural Science Foundation of China (32571110, 32070025, 62306333) and the Young Elite Scientists Sponsorship Program by CAST (2023QNRC001).

## Author contributions

W.C., J.X., and H.R. conceptualized the project. X.D.Z. and Y.X.Z. designed the PLGDL framework. X.D.Z. and Y.X.Z. prepared the benchmarks. Y.X.Z., D.L., M.Y.L., and M.L.L. evaluated the performance of methods over the benchmarks. X.W., Y.Y., X.F.Z., R.L., Y.L., Y.Z., and J.Z. conducted the immune evaluation experiment. X.D.Z. and Y.X.Z. wrote the original draft of the manuscript. All authors reviewed and edited the manuscript. J.X., H.R., and X.D.Z. supervised the project.

## Competing interests

The authors declare no competing interests.
