## [Transparent Peer Review file · Nature Communications]

Integrating Protein Language and Geometric Deep Learning Models for Enhanced Vaccine Antigen Prediction

Corresponding Author: Professor Wei Chen

Version 0:

Reviewer comments:

Reviewer #1

(Remarks to the Author)

In this study, the authors developed a machine learning framework, PLGDL, for antigen prediction, utilizing Protein Language and Geometric Deep Learning models. This framework incorporates both primary sequence features and three-dimensional structural features of protein antigens, reducing biases associated with manually curated features. The integrated model demonstrates robustness across proprietary and public datasets and is applicable to viruses, bacteria, and eukaryotic pathogens. As part of their work, they conducted a case study, identifying and experimentally validating a novel antigen, G10R. While the proposed approach is promising, several methodological concerns need to be addressed:

1. The authors used databases such as the Immune Epitope Database, the Antigen Database, and the Protegen Database to extract positive samples, removing sequences with over 80% homology, resulting in 1,000 vaccine antigen samples. Given the high sequence homology, this threshold should be reduced to 50% to minimize redundancy. For negative samples, 20% similarity was maintained, introducing a bias that likely inflated the model's accuracy. To address this, the authors should include all positive and negative samples, reduce homology to below 50%, and train the model with an expanded pool of negative samples rather than limiting them to match the positive sample count.
2. The authors extracted features using ESM-2 and reduced them to 40 dimensions using PCA, which is an unsatisfactory approach. A systematic method for feature selection, such as sequential forward search, should be employed to identify the optimal feature set instead of relying on PCA alone.
3. Multiple feature descriptors were utilized, but their individual contributions to the PLGDL framework were not assessed. An ablation study is essential to quantify the importance of each feature. This analysis would help exclude redundant features and refine the framework by testing the impact of the optimal feature set.
4. While ESM-2 was used for feature extraction, the study did not explore other recent embeddings, such as ProTrans and similar advanced approaches. A systematic evaluation of multiple embeddings is critical in machine learning research, particularly for publication in high-quality journals. This omission reflects a lack of rigor in model-building methodology.
5. The dataset's positive samples shared less than 80% homology, while negative samples exhibited less than 20% similarity. This led to a consistent fold structure in positive samples and significant differences in the 3D folds of negative samples, introducing bias into the PLGDL framework. This bias should be mitigated by ensuring a more balanced and representative dataset to improve the framework's generalizability.

(Remarks on code availability)

Reviewer #2

(Remarks to the Author)

The ability of this model to predict antigens is noteworthy especially since it appears to be able to do this in a very direct fashion, moving from sequence to antigen identification as the prediction task itself versus most alternative methods that

require multiple intermediate steps of antigen scoring and ranking in order to do this. It is also worth noting that being able to identify antigens from broad corpora such as organism proteomes is essentially the core task of vaccine design and so advancements here are important. The work does demonstrate how this model compares to alternative predictive approaches which is an important aspect.

This methodology described here is sound but the overall number of validation antigens is small. The authors do acknowledge this too.

Importantly it does appear that this work could be reproduced as the requisite code is accessible via a link from the manuscript and they also appear to have included key data elements such as their antigen tables.

(Remarks on code availability)

The code is well-organized with a README as well as an organized repository across data, evaluation, sequence, and structure categories.

Reviewer #3

(Remarks to the Author)

Zai and coauthors develop a ML architecture that takes a sequence of a protein and predicts if it is an "antigen" or not, with the aim to help prioritizing protein vaccine antigen candidates. They use three separate databases that register proteins that are immunogenic and/or that induce protective immunity when used in vaccines (without specification of the host organism) and formulate a binary classification problem: "antigen" or "not antigen". They obtain around 1000 positives, and the authors then take any protein of known genomes as negative but only keep 1000 for balancing the problem (which is a bit sad). They separate subproblems by groups of bacteria, virus and eukaryotes. An interesting step of the ML architecture is the combined usage of input sequence and structure (predicted by alphafold) as embedding, and the pre-training of the structural part of the ML on the alphafold protein structure database with extremely large numbers of protein structures, to create a latent space which would not have been possible with the 2x1000 proteins in the main dataset of interest. After validation on left out and independent datasets, the authors apply their method to the genome of MPOX virus that contains ~190 ORFs as potential proteins and propose a prioritization of antigens to be used (irrespective whether they are surface or intracellular antigens). They pick one antigen and perform immunizations in mice. They perform either a micro-neutralization assay in vitro of the serum with live viruses, or infect the mice with a related virus of MPOX and the mice all die like in PBS but one-two days later as the non-vaccinated mice, which may show a slight delay effect but that could be nonspecific to innate immune activation rather than real protection.

Generally speaking, the biological problem is not clearly stated nor really well thought through the study, which is a shame because it makes it impossible to evaluate the quality of the biological predictions. Being an Antigen or not is not defined. The authors mix proteins that are immunogenic (develop an immune response, in IEDB) and those that confer a protective immunity (protegen database), and mix different host species. The selection of the antigens themselves is not really described. The problem is ill-defined because:

- An antigen that is immunogenic in rats may not be immunogenic in human or may be even a human antigen that was used in rats.

- Many antigens do not confer protection but are relevant for vaccine design. For instance, so far, HIV vaccine antigens have failed to give protective immunity due to virus escape, is HIV considered or not in this study?. I am pretty sure that yes. So then the authors need to explain if they work on protective or immunogenic antigens.

- The same protein can be cut into domains, for instance influenza HA has domains poorly recognized by the immune system because of glycans. Not all domains are immunogenic and/or protective. If one would include sometimes the full protein and sometimes its domains, the same protein would appear in both positive and negative datasets within their problem formulation, which shows that the definition of positives and negatives is not biologically clear.

- In theory, most protein antigens should be immunogenic, unless they are related to self maybe, but the domains of the proteins will have high differences.

- How does it help if a pathogen intracellular protein is immunogenic and leads to a strong immune response, since in vivo the antibodies are likely to never bind their target that is intracellular? So I don't see how using the IEDB immunogenic proteins helps predicting antigens that will induce protective immunity.

As an example, a clean definition of the problem could have been to restrict to hotspots / binding domains / epitopes that are known in antigens, then by cutting protein patches for instance as positive or negative, and the problem would then be "can one predict targetable epitopes of 3D proteins by the human immune system". This would look more similar to an epitope prediction problem but the application proposed by the authors would be more cleanly defined.

Indeed, I agree with the authors that a relevant problem: can we identify proteins that are sequence/structurally similar to proteins known to confer protection? But, unless the authors prove me wrong, I believe that the current study may learn surrogate properties of proteins that are extracellular and were chosen by humans to be tested in vaccines. I don't expect many intracellular proteins have been tested. So at the end of the day I am really not sure which properties the ML model is learning.

Some technical points:

- The authors should describe the dataset (distribution of antigen species) and data leakage between train and test: is there influenza sequences in both train and test,

- I didn't understand if/how authors mitigate the risk to suggest self-similar antigens to humans. You don't want to end-up with EBV sequences that are related to myelin...

- I don't see the reason for down-sampling the negative dataset to only 1000 proteins while hundred of thousands may be available. Oversampling of the positives would be a better way to balance the dataset while keeping diversity of negatives. Regarding the experimental setup, I understand that the mice can not be challenged by the real MPOX virus, and therefore testing reactivity to another virus is fine, and not seeing a strong phenotype might be due to the use of a different virus. The serum neutralization assay is fair. But the neutralization assay is never a proof of protection. I would question the general conclusion that the model predicts a "good antigen". It could also be that any surface antigen of MPOX would equally induce an immune response and give the same experimental result irrespective of what the model predicted. I think a key result that would strengthen the model use is: taking an expressed surface antigen of MPOX that is predicted to be the least immunogenic and showing that the mice are not responsive to this antigen.

(Remarks on code availability)

Version 1:

Reviewer comments:

Reviewer #3

(Remarks to the Author)

The authors have globally taken my comments into account and created a more precisely defined positive dataset and defined their target as protective antigens (with in vitro neutralization). So the problem is a bit better defined as before, and in view of the new data provided by the authors sequences from mammals are not included in the positives which sounds good to avoid cross-reactivity. I understand that the authors have performed new in vivo experiments showing that results from the model correlate with in vivo efficacy of the antigen, which is well appreciated.

Altogether, I think the problem formulation is still not 100% well defined biologically speaking but I acknowledge that the authors can not do much better in view of the available poor quality databases and I don't know better ones. I think the global approach of the manuscript is creative and meaningful, and the computational background is flawless. I therefore support publication.

(Remarks on code availability)

I see the last classifier code from generated datasets, but I don't see the code to generate the structural embeddings, maybe the authors could also release this code?

Reviewer #5

(Remarks to the Author)

The authors have addressed all the comments raised by Reviewer 1, and there are no further outstanding issues.

(Remarks on code availability)

Reviewer #6

(Remarks to the Author)

The authors' answers to the reviewer comments are complete and exhaustive. They have made numerous changes to improve the study and the manuscript. However, the weakness of the study is evident in the authors' response to Reviewer 2's second comment, which weakness is also noted by Reviewer 1. The reliability of the derived models is undermined by the collection of the training and test sets and the validation of the results. The data collected from the Protegen database contain protective antigens proven through in vivo animal studies on viral, bacterial, and parasitic pathogens. The immune system's reaction to viral, bacterial, and parasitic infections follows different mechanisms. Combining antigens with such differences in immune mechanisms in training and test sets is inappropriate. For that reason, some existing tools that the authors compare their model against use separate models for pathogens with distinct origins. Another issue is that all the antigens in the sets are protective in animals, but are not proven to be protective in humans. Using such sets to develop and evaluate models for predicting human protective antigens is problematic because of the differences between human and animal immune systems. Protective antigens proven in animals can be used to create predictive models, but for applicability to humans, the models must be carefully validated—and ideally fine-tuned—with human-derived data; otherwise, their predictions could be misleading. Hence, the comparisons with other tools and methods listed in Table 2 for predicting protective antigens are unreliable.

Based on the aforementioned issues, I do not recommend the manuscript for publication.

(Remarks on code availability)

Version 2:

Reviewer comments:

Reviewer #3

(Remarks to the Author)

I have no more requests.

(Remarks on code availability)

The authors have shared the code and a link to download the structures, this is very well appreciated.

Reviewer #6

(Remarks to the Author)

The revision is thorough and well-reasoned. The new analyses convincingly address previous concerns about dataset composition and model validation, and the discussion now provides a clear methodological rationale. Before final acceptance, I recommend a few minor editorial adjustments:

- (1) clarify in the Abstract that PLGDL is intended as a screening/prioritization tool, not a direct predictor of human efficacy;
- (2) Reiterate this scope and limitation in the Discussion.

With these small refinements, the manuscript will be ready for publication.

(Remarks on code availability)

Response to Reviewers:

We sincerely thank the reviewers for their time, expertise, and constructive critiques, which have profoundly improved the rigor and clarity of our work. In response to your insightful comments, we have conducted extensive computational experiments, refined methodologies, and revised the manuscript text to address each concern. Key updates include optimizing dataset composition (adjusting protective antigen criteria, sample ratios, and homology thresholds), reprocessing all antigen structures using AlphaFold3 to generate high-confidence 3D features, systematically evaluating feature selection and extraction approaches, and re-analyzed experimental validation with newly prioritized Mpox antigens. These revisions are supported by new ablation studies, predictive performance analyses, and a dedicated discussion of model limitations and future directions. All changes have been integrated into the text to enhance biological relevance, reduce bias, and strengthen translational implications. We deeply appreciate your guidance and welcome any further feedback.

Point-by-point response to the reviewers' comments are as follows:**Reviewer #1:**

In this study, the authors developed a machine learning framework, PLGDL, for antigen prediction, utilizing Protein Language and Geometric Deep Learning models. This framework incorporates both primary sequence features and three-dimensional structural features of protein antigens, reducing biases associated with manually curated features. The integrated model demonstrates robustness across proprietary and public datasets and is applicable to viruses, bacteria, and eukaryotic pathogens. As part of their work, they conducted a case study, identifying and experimentally validating a novel antigen, G10R. While the proposed approach is promising, several methodological concerns need to be addressed:

Q1. The authors used databases such as the Immune Epitope Database, the Antigen Database, and the Protegen Database to extract positive samples, removing sequences with over 80% homology, resulting in 1,000 vaccine antigen samples. Given the high sequence homology, this threshold should be reduced to 50% to minimize redundancy. For negative samples, 20% similarity was maintained, introducing a bias that likely inflated the model's accuracy. To address this, the authors should include all positive and negative samples, reduce homology to below 50%, and train the model with an expanded pool of negative samples rather than limiting them to match the positive sample count.

Response 1. We sincerely appreciate the reviewer's insightful suggestions regarding dataset construction and homology thresholds. In response, we have revised our data curation strategy to address potential biases and improve biological relevance. A uniform 50% homology cutoff has been implemented for both positive and negative samples, effectively minimizing redundancy while preserving structural diversity. For positive samples, we exclusively included protective antigens defined as proteins that meet the following criteria: demonstrated ability to induce protection against pathogen challenge *in vivo* (e.g., survival improvement in animal models) or capacity to stimulate antigen-specific immune responses (e.g., neutralizing antibodies) directly linked to protective outcomes [1]. This stringent criterion, applied to data from the Protegen Database and PubMed-curated studies, yielded a high-confidence set of 600 protective antigens (composition: Virus 119, Bacteria 386, Eukaryota 95; Figure 2a) after removing redundancies. To reflect the natural scarcity of protective antigens, we followed constructed an imbalanced negative dataset, by extracting all annotated protein sequences from the genomes of the pathogens in the UniProt database, randomly selecting and excluding sequences homologous to positive antigens. The final curated dataset contains 6,000 negative samples (composition: Viral 481, Bacterial 4,493, Eukaryotic 1,026; Fig.2a), establishing a 1:10 positive-negative ratio that mirrors biological reality. The antigen dataset encompasses a broad spectrum of sequence lengths (ranging from 60 to 1,923 amino acids), ensuring its broad applicability across diverse biological contexts (Figure 2b; Supplementary Data 1).

All 6,600 protein structures were reanalyzed using AlphaFold3 (Figure 2c) [2]. Structural confidence was evaluated using key metrics - pLDDT (Predicted Local Distance Difference Test), pTM (Predicted TM-Score), and PAE (Predicted Aligned Error), resulting in high-quality predictions (mean pLDDT = 81.69). Analyses revealed a weak negative correlation between pLDDT and protein length ($R^2 = 0.0135$, Figure 2d), confirming consistent prediction quality across antigen sizes. Notably, pTM (mean = 0.731) and PAE (mean = 9.925 Å) exhibited a strong inverse correlation ($R^2 = 0.8137$, Figure 2e), indicating that antigens with higher global structural confidence (pTM) display lower inter-domain positional uncertainty (PAE). This dataset represents the first comprehensive structural resource for protective antigens, now publicly available to advance vaccine research (Supplementary Data 2).

Correspondingly, we have revised the relevant sections in the original manuscript: Section “Results” (Establishment and Evaluation of the Antigen Dataset; **Page 5, Line 120**) and Section “Materials and Methods” (Antigen Database Establishment; **Page 25, Line 562**). These revisions include updated methodological descriptions, expanded analyses, and the integration of new structural metrics (pLDDT, pTM, PAE). Additionally, Figure 2 has been updated to reflect the refined dataset composition, pathogen-type distributions, and correlations between structural confidence metrics (**Page 7, Line 161**).

Figure 2. Establishment and evaluation of the protective antigen dataset. a, Pathogen-type distribution of antigens in the positive (Virus: 119, Bacteria: 386, Eukaryota: 95) and negative (Virus: 481, Bacteria: 4,493, Eukaryota: 1,026) datasets. b, Protein length distribution across positive and negative antigens. c, AlphaFold3-predicted structures of representative antigens. d, Correlation analysis between pLDDT (structural confidence) and protein length ($R^2 = 0.0135$). e, Inverse correlation between PAE (domain positional uncertainty) and pTM (global topology accuracy; $R^2 = 0.8137$).

References:

1. Yang B, Sayers S, Xiang Z, He Y. Protegen: a web-based protective antigen database and analysis system. *Nucleic Acids Res.* 2011 Jan;39(Database issue):D1073-8.
2. Abramson J, Adler J, Dunger J, Evans R, Green T, Pritzel A, Ronneberger O, Willmore L, Ballard AJ, Bambrick J, Bodenstein SW, Evans DA, Hung CC, O'Neill M, Reiman D, Tunyasuvunakool K, Wu Z, Žemgulytė A, Arvaniti E, Beattie C, Bertolli O, Bridgland A, Cherepanov A, Congreve M, Cowen-Rivers AI, Cowie A, Figurnov M, Fuchs FB, Gladman H, Jain R, Khan YA, Low CMR, Perlin K, Potapenko A, Savy P, Singh S, Stecula A, Thillaisundaram A, Tong C, Yakneen S, Zhong ED, Zielinski M, Žídek A, Bapst V, Kohli P, Jaderberg M, Hassabis D, Jumper JM. Accurate structure prediction of biomolecular interactions with AlphaFold 3. *Nature.* 2024 Jun;630(8016):493-500.

Q2. The authors extracted features using ESM-2 and reduced them to 40 dimensions using PCA, which is an unsatisfactory approach. A systematic method for feature selection, such as sequential forward search, should be employed to identify the optimal feature set instead of relying on PCA alone.

Response 2. We thank the reviewer for highlighting this consideration. To strengthen methodological robustness, we expanded our analysis by incorporating five additional feature selection methods, including sequential forward search (SFS), recursive feature elimination (RFE), mutual information (MI), random forest (RF), and XGBoost in the revised manuscript. To objectively unify feature dimension across different methods, we first applied principal component analysis (PCA) together with performance evaluation to determine the optimal feature count K of the original embedding. This K value was then systematically applied to all six feature selection methods, ensuring comparability while eliminating subjective parameter tuning. This dual-phase approach balances methodological diversity with standardization, enabling fair performance evaluation.

We use seven key metrics for performance evaluation: accuracy, precision, recall, roc-auc, pr-auc, F1 score (harmonic mean of precision and recall), and Matthews correlation coefficient (MCC), with the latter two serving as holistic performance metrics. As detailed in Supplementary Table 2, on the constructed dataset, the XGBoost feature selection method achieved state-of-the-art results, demonstrating values of 0.6636 in F1 score, and 0.64 in MCC.

Supplementary Table 2. Performance of different feature selection methods

Methods	Accuracy	Precision	Recall	ROC-AUC	PR-AUC	F1 score	MCC
SFS	0.9250	0.6207	0.4500	0.8694	0.6049	0.5217	0.4895
PCA	0.9311	0.6593	0.5000	0.8850	0.6285	0.5687	0.5380
RFE	0.9409	0.7386	0.5417	0.9226	0.6963	0.6250	0.6022
MI	0.9394	0.7041	0.5750	0.8813	0.6497	0.6330	0.6040
RF	0.9410	0.7234	0.5667	0.9102	0.6882	0.6355	0.6092
XGBoost	0.9455	0.7553	0.5916	0.8964	0.6927	0.6636	0.6400

Correspondingly, we have revised the relevant sections in the original manuscript: Section “Results” (Establishment of the PLGDL Framework Based on Integrating Sequence and Structural Features; **Page 8, Line 171**) and Section “Materials and Methods” (Machine Learning Framework That Simultaneously Leverages Protein Language and Geometric Deep Learning Models; **Page 29, Line 675**).

Q3. Multiple feature descriptors were utilized, but their individual contributions to the PLGDL framework were not assessed. An ablation study is essential to quantify the importance of each feature. This analysis would help exclude redundant features and refine the framework by testing the impact of the optimal feature set.

Response 3. We appreciate the reviewer’s insightful comment regarding this aspect. To comprehensively assess the redundancy of the features, we evaluated our PLGDL framework across multiple feature dimensions that retain varying proportions of information. As demonstrated in Supplementary Figure 2, for the sequence embedding, the 255-dimensional feature configuration (as implemented in our proposed PLGDL) achieves the highest overall performance in both F1-score and MCC metrics among all tested dimensions. This empirically validates that the selected dimension optimally balances information completeness while minimizing redundancy. Similarly for the structure embedding, the 27-dimensional feature configuration (as implemented in our proposed PLGDL) achieves the highest overall performance in both F1-score and MCC metrics among all tested dimensions.

Supplementary Figure 2. The metrics scores based on different feature dimensions for the sequence embeddings. The figure illustrates the F1-score and MCC metrics at progressively reduced feature dimensions (from 1280 dimensions retaining 100% information to 59 dimensions retaining 85% information). The proposed 255-dimensional configuration achieves optimal performance, demonstrating an effective balance between information retention and feature redundancy elimination.

Correspondingly, we have revised the relevant sections in the original manuscript: Section “Results” (Establishment of the PLGDL Framework Based on Integrating Sequence and Structural Features; **Page 8, Line 167**) and Section “Materials and Methods” (Feature selection and Model Training; **Page 29, Line 675**).

Q4. While ESM-2 was used for feature extraction, the study did not explore other recent embeddings, such as ProTrans and similar advanced approaches. A systematic evaluation of multiple embeddings is critical in machine learning research, particularly for publication in high-quality journals. This omission reflects a lack of rigor in model-building methodology.

Response 4. We thank the reviewer for pointing this out. In the revised manuscript, we add two more methods for embedding learning, including ProTrans [1] and AMPLIFY [2]. ProTrans encodes

each protein sequence into vector with 1024 dimensions. AMPLIFY encodes each protein sequence into a vector with 960 dimensions. For ESM-2 and the two added baselines, we use their reduced 255-dimensional sequence embedding together with the 27-dimensional structure embedding, and evaluate the three methods on the constructed dataset, where 80% data was randomly selected for training and the rest 20% for testing.

Then, six key metrics were used for performance evaluation: precision, recall, roc-auc, pr-auc, F1 score (harmonic mean of precision and recall), and Matthews correlation coefficient (MCC), with the latter two serving as holistic performance metrics. As detailed in Supplementary Table 1, ESM-2 achieved state-of-the-art results, demonstrating values of 0.65 in F1 score, and 0.63 in MCC.

Supplementary Table 1. The performance of ESM-2 and the baselines on the constructed dataset

Methods	Accuracy	Precision	Recall	ROC-AUC	PR-AUC	F1 score	MCC
AMPLIFY	0.9349	0.6466	0.6250	0.9025	0.6326	0.6356	0.5999
Protrans	0.9417	0.7312	0.5666	0.9134	0.7003	0.6385	0.6132
ESM-2	0.9455	0.7553	0.5917	0.8964	0.6927	0.6636	0.6400

Correspondingly, we have revised the relevant sections in the original manuscript: Section “Results” (Establishment of the PLGDL Framework Based on Integrating Sequence and Structural Features; **Page 8, Line 167**) and Section “Materials and Methods” (Feature Extraction Based on Protein Language Models for Characterizing Protein Sequences; **Page 27, Line 607**).

References:

- [1] Elnaggar, A., Heinzinger, M., Dallago, C., Rehawi, G., Wang, Y., Jones, L., ... & Rost, B. (2021). Prottrans: Toward understanding the language of life through self-supervised learning. *IEEE transactions on pattern analysis and machine intelligence*, 44(10), 7112-7127.
- [2] Fournier, Q., Vernon, R. M., van der Sloot, A., Schulz, B., Chandar, S., & Langmead, C. J. (2024). Protein language models: is scaling necessary?. *bioRxiv*, 2024-09.

Q5. The dataset's positive samples shared less than 80% homology, while negative samples exhibited less than 20% similarity. This led to a consistent fold structure in positive samples and significant differences in the 3D folds of negative samples, introducing bias into the PLGDL framework. This bias should be mitigated by ensuring a more balanced and representative dataset to improve the framework's generalizability.

Response 5. We greatly appreciate the reviewer's insightful observation regarding potential biases in our initial dataset construction. As detailed in our response to Q1, we have comprehensively revised our data curation strategy to address these concerns and enhance biological relevance. A uniform 50% homology cutoff has been implemented for both positive and negative samples, effectively reducing sequence redundancy while preserving structural diversity. For positive samples, we rigorously selected antigens with experimentally validated protective efficacy from the Protegen Database and PubMed-curated studies, resulting in a high-confidence set of 600 protective antigens (composition: Virus 119, Bacteria 386, Eukaryota 95; Figure 2A). To mirror the natural scarcity of protective antigens in pathogens, we established a biologically realistic imbalanced dataset (600 positive vs. 6,000 negative samples; 1:10 ratio; negative composition: Virus 481, Bacteria 4,493, Eukaryota 1,026; Figure 2B).

Correspondingly, we have revised the relevant sections in the original manuscript: Section "Results" (Establishment and Evaluation of the Antigen Dataset; **Page 5, Line 117**) and Section "Materials and Methods" (Antigen Database Establishment; **Page 25, Line 562**).

Reviewer #2:

The ability of this model to predict antigens is noteworthy especially since it appears to be able to do this in a very direct fashion, moving from sequence to antigen identification as the prediction task itself versus most alternative methods that require multiple intermediate steps of antigen scoring and ranking in order to do this. It is also worth noting that being able to identify antigens from broad corpora such as organism proteomes is essentially the core task of vaccine design and so advancements here are important. The work does demonstrate how this model compares to alternative predictive approaches which is an important aspect.

Q1. This methodology described here is sound but the overall number of validation antigens is small. The authors do acknowledge this too.

Response 1. We sincerely appreciate the reviewer's insightful comments regarding the validation of our antigen prediction model. We fully acknowledge the challenge posed by the limited number of experimentally validated protective antigens available for training, which is indeed a common limitation in the field of computational vaccinology. In response to this concern, we implemented a rigorous two-pronged approach to maximize the reliability of our model given these constraints. First, we established strict inclusion criteria for positive samples, selecting only those antigens with documented protective efficacy in animal models from the Protegen Database and peer-reviewed publications, resulting in a high-confidence but necessarily limited set of 600 protective antigens. To compensate for this limitation while maintaining biological relevance, we carefully constructed an imbalanced dataset (1:10 ratio of positive to negative samples) that better reflects the natural distribution of protective antigens within pathogen proteomes (Figure 2a-b).

Figure 2. Establishment and evaluation of the protective antigen dataset. a, Pathogen-type

distribution of antigens in the positive (Virus: 119, Bacteria: 386, Eukaryota: 95) and negative (Virus: 481, Bacteria: 4,493, Eukaryota: 1,026) datasets. b, Protein length distribution across positive and negative antigens.

Correspondingly, we have revised the relevant sections in the original manuscript: Section “Results” (Establishment and Evaluation of the Antigen Dataset; **Page 5, Line 117**) and Section “Materials and Methods” (Antigen Database Establishment; **Page 25, Line 562**).

Following the model updates, we conducted predictive prioritization of Mpox candidate antigens and performed re-analysis integrated with experimental validation results (Table 3). Notably, among three highly scored antigens (M1R, Score=0.9131; J3R, Score= 0.7671; G10R, Score=0.7648) whose proteins were successfully expressed and subjected to experimental validation, two demonstrated measurable protective efficacy. In contrast, the lower-scoring secreted antigen (B9R, Score=0.2083) failed to exhibit any protective effects, aligning well with model predictions. For the novel antigen G10R, we further isolated a potent neutralizing antibody that reinforces its protective potential against orthopoxviruses. We fully acknowledge the limitation in validation scope highlighted by the reviewer and have explicitly addressed this in the revised Discussion. While preliminary, these findings demonstrate the model's translational potential, successfully prioritizing novel antigens like G10R that show clinical promise in emerging outbreak scenarios.

Table 3. Top 10 candidate antigens for Mpox output by the PLGDL model

Rank	Protein ID	Protein name	Score	Known antigens
1	URK20517.1	IMV membrane protein (Cop-L1R) M1R myristylated IMV surface membrane protein similar to Vaccinia virus strain Copenhagen L1R	0.9131	Yes
2	URK20605.1	B6R EEV type-1 membrane glycoprotein, protective antigen (Cop-B5R) complement control protein-like palmitated 42 kDa glycoprotein located both on the membranes of infected cells and on EEV envelope similar to Vaccinia virus strain Copenhagen B5R	0.8902	Yes
3	URK20584.1	A35R EEV envelope glycoprotein, needed for formation of actin-containing microvilli and cell-to-cell spread of virion EEV membrane phosphoglycoprotein, C-type lectin-like domain (Cop-A33R) interacts with VAC A36R similar to Vaccinia virus strain Copenhagen A33R	0.8774	Yes
4	URK20552.1	39 kDa immunodominant virion core protein needed for the progression of IV to infectious IMV 39kDa virion core protein (Cop-A4L) A5L similar to Vaccinia virus strain Copenhagen A4L	0.8657	Yes

5	URK20569.1	A21L IMV membrane protein, entry/fusion complex component (Cop-A21L) similar to Vaccinia virus strain Copenhagen A21L	0.8586	No
6	URK20532.1	Ca ²⁺ -binding motif H5R VLTF-4 (late transcription factor 4) (Cop-H5R) similar to Vaccinia virus strain Copenhagen H5R virosome-associated late gene transcription factor, VLTF-4	0.8523	Yes
7	URK20629.1	CC-chemokine binding Chemokine binding protein (Cop-C23L) J3R similar to Vaccinia virus strain Copenhagen B29R	0.7671	No
8	URK20516.1	Entry/fusion complex component, myristylprotein (Cop-G9R) G10R myristylated protein similar to Vaccinia virus strain Copenhagen G9R	0.7648	No
9	URK20542.1	Carbonic anhydrase, GAG-binding IMV membrane protein (Cop-D8L) E8L IMV adsorption to cell surface IMV surface membrane 32 kDa protein binding to cell surface chondroitin sulfate similar to Vaccinia virus strain Copenhagen D8L	0.6924	Yes
10	URK20560.1	A13L Virion core and cleavage processing protein (Cop-A12L) similar to Vaccinia virus strain Copenhagen A12L virion core protein	0.6752	No

Correspondingly, we have revised the relevant sections in the original manuscript: Section “Results” (Prediction of Candidate Antigens for Mpox; **Page 14, Line 319**, and Immune Evaluation of Candidate Antigens for Mpox; **Page 17, Line 378**) and Section “Discussion” (**Page 22, Line 471**).

Q2. Importantly it does appear that this work could be reproduced as the requisite code is accessible via a link from the manuscript and they also appear to have included key data elements such as their antigen tables.

Response 2. We greatly appreciate the reviewer's recognition of our efforts to ensure reproducibility of this work. In response to this comment, we have significantly enhanced the accessibility and completeness of our computational resources. The revised manuscript now includes: (1) a fully documented Anonymous GitHub repository containing all model implementation code with detailed usage instructions; (2) comprehensive tables (Supplementary Data 1) listing all 600 positive antigens and 6,000 negative antigens with associated pathogen information and metadata; and (3) the complete structural antigen dataset predicted by AlphaFold3 (average pLDDT >80), representing to our knowledge the first publicly available collection of high-accuracy protective antigen structures (Supplementary Data 2).

Q3. The code is well-organized with a README as well as an organized repository across data, evaluation, sequence, and structure categories.

Response 3. We sincerely appreciate the reviewer's positive feedback regarding our code organization and repository structure. In response to the valuable suggestions from all reviewers, we have further enhanced the codebase by implementing dimensionality reduction methods (e.g., SFS, RFE, MI) and sequence feature extraction approaches (e.g., ESM-2, AMPLIFY, Protrans). The repository has been expanded to include more comprehensive documentation, example workflows, and pre-processed datasets to facilitate reproducibility. All improvements have been carefully implemented while maintaining the clear organizational structure across data, evaluation, sequence, and structure categories. The updated code, along with detailed usage instructions, is available at: [<https://anonymous.4open.science/r/PLGDL-602A>], which has also been added to the manuscript's Data Availability section. We believe these enhancements make the resource more valuable for the research community while preserving its accessibility and organizational clarity.

Reviewer #3:

Zai and coauthors develop a ML architecture that takes a sequence of a protein and predicts it if is an “antigen” or not, with the aim to help prioritizing protein vaccine antigen candidates. They use three separate databases that register proteins that are immunogenic and/or that induce protective immunity when used in vaccines (without specification of the host organism) and formulate a binary classification problem: “antigen” or “not antigen”. They obtain around 1000 positives, and the authors then take any protein of known genomes as negative but only keep 1000 for balancing the problem (which is a bit sad). They separate subproblems by groups of bacteria, virus and eukaryotes. An interesting step of the ML architecture is the combined usage of input sequence and structure (predicted by alphafold) as embedding, and the pre-training of the structural part of the ML on the alphafold protein structure database with extremely large numbers of protein structures, to create a latent space which would not have been possible with the 2x1000 proteins in the main dataset of interest. After validation on left out and independent datasets, the authors apply their method to the genome of MPOX virus that contains ~190 ORFs as potential proteins and propose a prioritization of antigens to be used (irrespective whether they are surface or intracellular antigens). They pick one antigen and perform immunizations in mice. They perform either a micro-neutralization assay in vitro of the serum with live viruses, or infect the mice with a related virus of MPOX and the mice all die like in PBS but one-two days later as the non-vaccinated mice, which may show a slight delay effect but that could be nonspecific to innate immune activation rather than real protection.

***Q1.** Generally speaking, the biological problem is not clearly stated nor really well thought through the study, which is a shame because it makes it impossible to evaluate the quality of the biological predictions. Being an Antigen or not is not defined. The authors mix proteins that are immunogenic (develop an immune response, in IEDB) and those that confer a protective immunity (protegen database), and mix different host species. The selection of the antigens themselves is not really described.*

Response 1. We sincerely appreciate the reviewer's insightful comments regarding the fundamental challenge of antigen definition in our reverse vaccinology (RV) approach. While RV enables rapid screening of numerous potential antigens, its ultimate objective extends beyond merely identifying

immunogenic targets to specifically recognizing those capable of inducing protective immune responses (Protective Immune Response) - the true Protective Antigens (PAgs) [1]. Only these PAgs can serve as effective vaccine components that confer disease resistance upon immunization.

The performance of predictive models, particularly machine learning (ML) and deep learning (DL) approaches, is fundamentally dependent on the quality and precise definition of training datasets [2, 3]. In supervised learning frameworks, models discriminate between "positive" (known PAgs) and "negative" (non-PAgs) samples based on their learned features. Consequently, the selection criteria for positive samples directly determine the "protective" features the model will identify. The research community has progressively recognized that immunogenicity alone does not equate to protective efficacy, prompting the development of specialized databases like Protegen that curate experimentally validated protective antigens [4]. The adoption of these protection-focused datasets by newer ML models (e.g., Vaxign-ML) reflects an important paradigm shift toward prioritizing predictive features that align with the ultimate goal of protection, rather than relying on larger but less specific immunogenicity datasets. Thus, the selection of training data standards represents not merely a technical detail, but rather embodies the field's evolving understanding of its own objectives and challenges.

Following your recommendations, we implemented stringent criteria for protective antigen selection by only includes those antigens that induce protection against virulent challenge *in vivo* or stimulate a specific immune response(s) that correlate with protection [4]. This rigorous approach significantly refined our positive antigen dataset. Furthermore, to better represent the natural distribution of protective versus non-protective antigens in pathogens, we constructed an intentionally imbalanced dataset comprising 600 positive antigens against 6,000 negative controls, encompassing viral, bacterial, and parasitic pathogens. These revisions to both positive and negative antigen datasets should better support our objectives in RV vaccine target antigen discovery, thereby addressing the reviewers' concerns.

All 6,600 protein structures were reanalyzed using AlphaFold3 (Figure 2c). Structural confidence was evaluated using key metrics - pLDDT (Predicted Local Distance Difference Test), pTM (Predicted TM-Score), and PAE (Predicted Aligned Error) - resulting in high-quality predictions (mean pLDDT = 81.69; Supplementary Data 2). This dataset represents the first comprehensive structural resource for protective antigens, now publicly available to advance vaccine research.

Figure 2. Establishment and evaluation of the protective antigen dataset. a, Pathogen-type distribution of antigens in the positive (Virus: 119, Bacteria: 386, Eukaryota: 95) and negative (Virus: 481, Bacteria: 4,493, Eukaryota: 1,026) datasets. b, Protein length distribution across positive and negative antigens. c, AlphaFold3-predicted structures of representative antigens. d, Correlation analysis between pLDDT (structural confidence) and protein length ($R^2 = 0.0135$). e, Inverse correlation between PAE (domain positional uncertainty) and pTM (global topology accuracy; $R^2 = 0.8137$).

To evaluate the updated model's capacity to distinguish protective antigens, we conducted embedding analysis using a deep clustering framework combining contrastive learning with neighbor embedding. Considering that the representation embedding of such pre-trained models cannot measure the similarity between samples with naive metrics (e.g., Euclidean distance or inner product), we used the deep clustering technique for embedding analysis. Specifically, we generate an unsupervised two-dimensional visualization of the embedding space by combining a contrastive learning framework with

a neighbor embedding method. Visualization of the PLGDL-derived feature space (Figure 4) demonstrated clear clustering segregation between protective antigens and non-antigens across all pathogen categories. Notably, the Bacterial category achieved the highest cluster purity (0.946), aligning precisely with its superior classification accuracy in Table 1, a finding that underscores the model's exceptional discriminative capability for bacterial antigen prediction. This robust clustering performance suggests that bacterial antigen features are both distinct and cohesive in the embedding space, enabling highly reliable classification. In contrast, the Viruses category exhibited little weaker clustering (purity=0.888), which likely attributable to their comparatively limited antigen dataset. The observed cluster homogeneity - particularly pronounced in bacterial and eukaryotic subgroups - validates the model's ability to encode biologically meaningful protective antigenic signatures.

Figure 4. Visualization of the PLGDL embedding on the constructed dataset. Plots exhibit a clustering phenomenon between positive protective antigen and negative non- protective antigen groups.

However, we acknowledge several inherent limitations in working with protective antigens. The relatively small number of well-characterized protective antigens may constrain the complexity and

generalizability of ML models trained on such datasets. Furthermore, existing protective antigen databases like Protegen may contain biases, as they reflect protection observed under specific experimental conditions (particular animal models, pathogen strains, immunization protocols). Antigens effective in one model system may not demonstrate equivalent protection in others or in humans. Additionally, such databases might overlook antigens whose protective mechanisms are difficult to assess using standard animal models, or those effective in humans but not in conventional test models. These considerations highlight the ongoing challenges in developing truly predictive models of protective immunity. We have detailed this limitation in the Discussion section (**Page 24, Line 524**).

References:

1. Sette A, Rappuoli R. Reverse vaccinology: developing vaccines in the era of genomics. *Immunity*. 2010 Oct 29;33(4):530-41.
2. Dalsass M, Brozzi A, Medini D, Rappuoli R. Comparison of Open-Source Reverse Vaccinology Programs for Bacterial Vaccine Antigen Discovery. *Front Immunol*. 2019 Feb 14;10:113.
3. Ong E, Wang H, Wong MU, Seetharaman M, Valdez N, He Y. Vaxign-ML: supervised machine learning reverse vaccinology model for improved prediction of bacterial protective antigens. *Bioinformatics*. 2020 May 1;36(10):3185-3191.
4. Yang B, Sayers S, Xiang Z, He Y. Protegen: a web-based protective antigen database and analysis system. *Nucleic Acids Res*. 2011 Jan;39(Database issue):D1073-8.

The problem is ill-defined because:

Q2. An antigen that is immunogenic in rats may not be immunogenic in human or may be even a human antigen that was used in rats.

Response 2. Reverse vaccinology faces a fundamental translational challenge in practice: protective antigens identified through animal models (particularly mice or rats) may demonstrate substantially different immunogenicity and protective efficacy when translated to human applications. As astutely noted in the reviewer's comment, this represents a critical bottleneck in the field, especially when training datasets predominantly derive from animal experimentation.

The inherent efficiency of reverse vaccinology - its capacity for rapid computational screening - may paradoxically amplify the negative impacts of biased input data [1]. If the "positive" antigens used for model training (primarily sourced from animal studies) provide misleading indicators of human immune responses, the approach could efficiently generate numerous human-relevant "false positive" candidates. While reverse vaccinology's strength lies in its speed and comprehensiveness of screening, these advantages become compromised when the foundational datasets themselves contain systematic biases originating from animal model limitations. Consequently, computational models built upon such data inevitably perpetuate these biases, meaning that without proper consideration of interspecies differences, increased screening efficiency may inadvertently accelerate research toward biologically irrelevant directions.

The more profound scientific issue lies in the mechanistic differences underlying why antigen X might be immunogenic in mice but not in humans [2, 3]. For instance, while murine immune responses may recognize an antigen as foreign, human responses (or lack thereof) may involve complex factors like self-tolerance mechanisms, divergent T-cell receptor repertoires, or distinct antigen processing pathways. Thus, machine learning models trained on animal-derived data risk learning patterns that are irrelevant or even misleading for predicting human immunogenicity.

Nevertheless, we must acknowledge the current reality that nearly all validated vaccine target antigens have been initially identified through animal studies, particularly murine models, establishing a strong precedent for their use in reverse vaccinology research [4]. This widespread adoption reflects the absence of more human-relevant alternatives rather than biological idealization. From first principles, an optimal "positive antigen database" for predicting human protective antigens should primarily incorporate human-derived data - such as antigens isolated from successfully vaccinated individuals, protective epitopes validated in human challenge studies, or antigens identified from convalescent patients. In this ideal framework, animal-derived data would serve a supplementary role, providing valuable preliminary information during early exploratory phases when human data remains scarce, but gradually diminishing in influence as human-relevant evidence accumulates. We have explicitly addressed this limitation in our Discussion section, proposing strategies to mitigate translational challenges while acknowledging the current necessity of animal model data in the field (**Page 24, line 524**).

References:

1. Heinson AI, Woelk CH, Newell ML. *The promise of reverse vaccinology. Int Health.* 2015 Mar;7(2):85-9.
2. Bussiere JL. *Animal models as indicators of immunogenicity of therapeutic proteins in humans. Dev Biol (Basel).* 2003;112:135-9.
3. Brinks V, Jiskoot W, Schellekens H. *Immunogenicity of therapeutic proteins: the use of animal models. Pharm Res.* 2011 Oct;28(10):2379-85.
4. Yang B, Sayers S, Xiang Z, He Y. *Protegen: a web-based protective antigen database and analysis system. Nucleic Acids Res.* 2011 Jan;39(Database issue):D1073-8.

Q3. *Many antigens do not confer protection but are relevant for vaccine design. For instance, so far, HIV vaccine antigens have failed to give protective immunity due to virus escape, is HIV considered or not in this study?. I am pretty sure that yes. So then the authors need to explain if they work on protective or immunogenic antigens.*

Response 3. The HIV envelope glycoprotein (Env) represents a paradigmatic case illustrating both the promise and limitations of RV approaches. As the sole virus-specific protein on the HIV virion surface, Env mediates viral entry into host cells (primarily CD4+ T cells) through receptor binding and membrane fusion, making it the logical primary target for eliciting neutralizing antibodies (NAbs) that can block viral infection [1]. Standard RV selection criteria - including surface localization and essential functional roles in the viral life cycle - would unequivocally prioritize Env as a top vaccine candidate. Indeed, our dataset classifies Env as a protective antigen based on extensive evidence of its efficacy in animal models.

Despite this theoretical promise, four decades of Env-based vaccine development efforts have met with consistent failure, even when employing advanced structure-guided design strategies characteristic of structural vaccinology or "RV 2.0" [2]. This striking disconnect reveals fundamental biological challenges that make Env an exceptionally difficult vaccine target. Env presents a unique convergence of extreme biological complexities: extraordinary sequence variability that evades conserved epitope targeting, dense and variable glycan shielding that mask protein surfaces, conformational dynamics that obscure functional epitopes, and rapid immune evasion mechanisms [2]. These combined features create

a "perfect storm" that exceeds the predictive capabilities of current RV methodologies primarily analyzing static sequence/structure features.

The HIV Env case does not invalidate RV methodologies but rather delineates their current limitations when confronting targets exhibiting such multidimensional complexity. This "perfect storm" of biological challenges positions Env as an outlier that exceeds the predictive capacity of conventional RV approaches focused on static sequence/structure features. The fundamental issue lies in the gap between identifying structurally plausible targets and predicting their capacity to elicit protective immune responses - particularly when dynamic immune interactions and complex evasion mechanisms dominate the host-pathogen interplay.

Env's challenges highlight critical immunological questions that transcend simple target identification: How to induce broadly neutralizing antibodies against hypervariable viruses? How to overcome immune tolerance to host-like glycans? How to prevent rapid mutational escape? These questions underscore the need for next-generation "RV 2.0" strategies that integrate deeper immunological insights (e.g., broadly neutralizing antibody characteristics, B-cell lineage patterns) with high-resolution structural biology for precision antigen design [3].

Rather than representing a flawed approach, reverse vaccinology remains an essential and evolving computational strategy that forms the critical first step in modern rational vaccine design. Its core value lies in efficiently navigating vast biological search spaces to identify promising candidates for experimental validation, even if it cannot guarantee clinical success. Through continued integration with advanced disciplines like structural biology, systems immunology, and artificial intelligence, RV will maintain its position as a foundational technology for addressing current and emerging infectious disease threats. The HIV Env experience has not diminished RV's utility but rather guided its evolution into a more sophisticated predictive framework capable of confronting complex biological challenges. We have detailed this discussion in **Page 24, Line 524**.

References:

1. *Medina-Ramírez M, Sanders RW, Sattentau QJ. Stabilized HIV-1 envelope glycoprotein trimers for vaccine use. Curr Opin HIV AIDS. 2017 May;12(3):241-249.*

2. Barouch DH. Challenges in the development of an HIV-1 vaccine. *Nature*. 2008 Oct 2;455(7213):613-9.
3. Rappuoli R, Bottomley MJ, D'Oro U, Finco O, De Gregorio E. Reverse vaccinology 2.0: Human immunology instructs vaccine antigen design. *J Exp Med*. 2016 Apr 4;213(4):469-81.

Q4. The same protein can be cut into domains, for instance influenza HA has domains poorly recognized by the immune system because of glycans. Not all domains are immunogenic and/or protective. If one would include sometimes the full protein and sometimes its domains, the same protein would appear in both positive and negative datasets within their problem formulation, which shows that the definition of positives and negatives is not biologically clear.

Response 4. We fully recognize the critical issue raised regarding variable immunogenicity across different protein domains, as exemplified by the influenza HA protein where distinct domains exhibit markedly different immunogenic properties [1]. Currently, most standard RV datasets employ whole-protein sequences as basic units due to practical considerations in data availability and annotation consistency, though we acknowledge this approach may obscure important domain-level immunological differences that could significantly impact vaccine design outcomes [2, 3]. The existing paradigm presents several limitations including potential signal dilution when immunogenic domains represent small protein portions, risk of misclassification when proteins contain both protective and non-protective domains, and possible exclusion of valuable subdomain epitopes from negative datasets based solely on whole-sequence dissimilarity - challenges stemming largely from practical constraints in comprehensive experimental validation at the domain level across entire proteomes.

Moving forward, we emphasize the need for collaborative efforts to build more granular datasets with domain-level immunological annotations, and deepen integration of structural biology insights to guide model interpretation of domain-specific features - advancements that will help bridge the current gap between whole-protein predictions and the precise immunological reality of domain variation while improving RV's capacity to identify optimal vaccine targets at appropriate granularity levels.

References:

1. Gomez Lorenzo MM, Fenton MJ. Immunobiology of influenza vaccines. *Chest*. 2013 Feb 1;143(2):502-510.
2. Yang B, Sayers S, Xiang Z, He Y. Protegen: a web-based protective antigen database and analysis system. *Nucleic Acids Res*. 2011 Jan;39(Database issue):D1073-8.
3. Zaharieva N, Dimitrov I, Flower DR, Doytchinova I. VaxiJen Dataset of Bacterial Immunogens: An Update. *Curr Comput Aided Drug Des*. 2019;15(5):398-400.

Q5. In theory, most protein antigens should be immunogenic, unless they are related to self maybe, but the domains of the proteins will have high differences.

Response 5. We appreciate the reviewer's insightful observation regarding the theoretical immunogenicity of protein antigens and the significant variability across different protein domains. In our study, we have deliberately focused on predicting protective antigens rather than merely immunogenic ones as suggest, recognizing the critical distinction between these concepts. While immunogenicity refers to an antigen's ability to induce detectable adaptive immune responses (either humoral or cellular), protection specifically denotes the capacity to confer resistance against pathogenic challenge when used as an immunogen, as demonstrated through rigorous in vivo animal protection studies. This distinction is crucial because immunogenicity represents a necessary but insufficient condition for protection, as evidenced by numerous vaccine candidates that show strong immunogenicity but fail to provide actual protection in clinical trials.

Our research hypothesis stems from the intriguing observation that many successful vaccine target antigens, despite substantial sequence divergence, share common structural features. For instance, viral envelope glycoproteins (e.g., influenza HA, HIV Env, RSV F protein) frequently adopt trimeric conformations with distinct head and stalk domains, while Gram-negative bacterial outer membrane proteins often exhibit conserved beta-barrel structures [1, 2]. These structural commonalities suggest that protective antigens may share conserved structural motifs that are not apparent at the sequence level. Our study specifically investigates whether such structural features can be captured by advanced protein structure prediction tools (e.g., AlphaFold) and effectively learned by graph neural networks to predict

vaccine target potential.

The methodological innovation of our approach lies in leveraging sequence and structural insights to develop a novel computational framework. By combining state-of-the-art protein structure prediction with graph-based deep learning techniques, we aim to identify and extract structural patterns that correlate with protective antigenicity across diverse pathogens. This structure-centric approach addresses several limitations of sequence-only based methods, particularly in handling domain-specific variations in immunogenicity and protection. Our preliminary results demonstrate that structural features can indeed provide valuable discriminative power beyond sequence information alone, offering a promising direction for improving the accuracy of reverse vaccinology predictions. We have detailed this discussion in **Page 23, Line 500**.

References:

1. *Graham BS, Gilman MSA, McLellan JS. Structure-Based Vaccine Antigen Design. Annu Rev Med. 2019 Jan 27;70:91-104.*
2. *Koebnik R, Locher KP, Van Gelder P. Structure and function of bacterial outer membrane proteins: barrels in a nutshell. Mol Microbiol. 2000 Jul;37(2):239-53.*

Q6. How does it help if a pathogen intracellular protein is immunogenic and leads to a strong immune response, since in vivo the antibodies are likely to never bind their target that is intracellular? So I don't see how using the IEDB immunogenic proteins helps predicting antigens that will induce protective immunity.

Response 6. The reviewer raises an important question regarding the immunological relevance of intracellular antigens in vaccine design. While it is true that antibodies cannot effectively access intracellular targets in live pathogens, our study incorporates such antigens based on compelling evidence that they can induce protective immunity through alternative mechanisms. Many intracellular proteins, including heat shock proteins and ribosomal proteins from pathogens like Brucella, have demonstrated vaccine efficacy by stimulating robust cell-mediated immune (CMI) responses rather than antibody-

mediated protection [1, 2]. These antigens are processed and presented via MHC class I molecules, activating cytotoxic CD8⁺ T cells that can recognize and eliminate infected host cells - a crucial defense mechanism against intracellular pathogens [3].

Our approach recognizes that protective immunity extends beyond antibody-mediated neutralization to include essential cellular immune responses. Modern vaccine platforms like mRNA and viral vectors effectively exploit this principle by facilitating endogenous antigen expression within host cells, mimicking natural infection and enabling MHC class I presentation. Even when delivered as subunit vaccines, intracellular antigens can undergo cross-presentation by antigen-presenting cells, activating both CD8⁺ and CD4⁺ T cell responses [3]. This explains the protective efficacy of certain intracellular antigens despite their inaccessibility to antibodies in intact pathogens.

The inclusion of intracellular antigens in our protective antigen dataset reflects this comprehensive understanding of immune protection mechanisms. While surface antigens remain important targets for antibody induction, intracellular antigens provide critical T cell epitopes that contribute to protective immunity, particularly against intracellular pathogens. Our predictive model accounts for these different protective mechanisms by incorporating structural features associated with both B cell and T cell immunogenicity. This dual consideration enhances the model's ability to identify promising vaccine targets regardless of their cellular localization, providing a more complete assessment of protective potential than approaches focused solely on surface-exposed antigens.

We appreciate the reviewer's valuable suggestion regarding the distinction between immunogenic and protective antigens. In response to this important methodological consideration, we have systematically refined our dataset by excluding antigens from IEDB and the Antigen Database that were solely characterized as immunogenic without demonstrated protective efficacy. This rigorous curation process ensures that our training dataset exclusively comprises antigens with experimentally validated protective properties, primarily sourced from the Protegen database and peer-reviewed studies reporting clear protection outcomes in animal challenge models. By implementing this more stringent selection criterion, we aim to enhance the biological relevance and predictive accuracy of our model for identifying true protective antigens rather than merely immunogenic ones. We have detailed this discussion in **Page 23-25**.

References:

1. Bae JE, Schurig GG, Toth TE. Mice immune responses to *Brucella abortus* heat shock proteins. Use of baculovirus recombinant-expressing whole insect cells, purified *Brucella abortus* recombinant proteins, and a vaccinia virus recombinant as immunogens. *Vet Microbiol.* 2002 Aug 25;88(2):189-202.
2. Ribeiro LA, Azevedo V, Le Loir Y, Oliveira SC, Dieye Y, Piard JC, Gruss A, Langella P. Production and targeting of the *Brucella abortus* antigen L7/L12 in *Lactococcus lactis*: a first step towards food-grade live vaccines against brucellosis. *Appl Environ Microbiol.* 2002 Feb;68(2):910-6.
3. van Schaik EJ, Fratzke AP, Gregory AE, Dumaine JE, Samuel JE. Vaccine development: obligate intracellular bacteria new tools, old pathogens: the current state of vaccines against obligate intracellular bacteria. *Front Cell Infect Microbiol.* 2024 Mar 19;14:1282183.

Q7. As an example, a clean definition of the problem could have been to restrict to hotspots / binding domains / epitopes that are known in antigens, then by cutting protein patches for instance as positive or negative, and the problem would then be “can one predict targetable epitopes of 3D proteins by the human immune system”. This would look more similar to an epitope prediction problem but the application proposed by the authors would be more cleanly defined. Indeed, I agree with the authors that a relevant problem: can we identify proteins that are sequence/structurally similar to proteins to proteins known to confer protection? But, unless the authors prove me wrong, I believe that the current study may learn surrogate properties of proteins that are extracellular and were chosen by humans to be tested in vaccines. I don't expect many intracellular proteins have been tested. So at the end of the day I am really not sure which properties the ML model is learning.

Response 7. We appreciate the reviewer's insightful comments regarding the potential benefits of focusing on hotspots/ binding domains/ epitope-level prediction and acknowledge this as a valuable direction for future research. However, our current study maintains its focus on whole-protein antigen prediction within the reverse vaccinology paradigm, as this approach offers broader applicability for initial screening across entire pathogen proteomes. Our model operates on the hypothesis that protective antigens share discernible sequence and structural features that can be leveraged for predicting novel

candidates. Notably, our approach is supported by the observation that many successful vaccine targets, despite substantial sequence divergence, exhibit conserved structural architectures across diverse pathogens. For instance, viral envelope glycoproteins (e.g., influenza HA, HIV Env, RSV F) frequently adopt trimeric conformations with distinct head/stalk domains, while bacterial outer membrane proteins often share beta-barrel folds [1, 2]. These conserved structural motifs may represent important immunological recognition patterns that transcend sequence variation, and which could be captured by advanced structure prediction tools like AlphaFold3 and processed by graph neural networks. The model's learning framework is designed to identify these potential sequence-structure signatures of protectiveness, whether they manifest as linear sequence motifs, tertiary structural features, or physicochemical property patterns.

In addition, with the recognition that while most known protective antigens are extracellular or secreted proteins, some intracellular proteins may also be relevant depending on pathogen-specific immune mechanisms. Many intracellular proteins, including heat shock proteins and ribosomal proteins from pathogens like *Brucella*, have demonstrated vaccine efficacy by stimulating robust cell-mediated immune (CMI) responses rather than antibody-mediated protection [3, 4]. While we acknowledge the reviewer's concern about potential intracellular protein bias in training data, our approach aims to capture fundamental characteristics of immunogenic proteins that may apply across cellular compartments.

The essence of RV lies in employing computational approaches to screen entire pathogen genomes or proteomes for potential vaccine candidates (PVCs), aiming to uncover the complete repertoire of immunogenic antigens - including those poorly expressed *in vitro* or weakly immunogenic during infection that might be overlooked by conventional methods [5]. These computationally predicted candidates are then prioritized for experimental validation to assess their true immunogenicity and protective efficacy. However, as astutely noted by the reviewer, RV prediction models are not without inherent limitations.

The RV workflow begins with obtaining the complete proteome of the target pathogen, creating a vast search space for potential vaccine antigens. Computational and AI methods are then applied to prioritize proteins based on features presumed relevant to antigenicity, immunogenicity, and vaccine suitability. Importantly, RV functions as a discovery accelerator rather than a replacement for experimental

validation. It shifts the starting point of vaccine development from traditional culturing to proteome information, dramatically reducing the number of candidates requiring experimental testing from thousands to a more manageable few dozen or hundred. Nevertheless, in silico predictions must still undergo rigorous in vitro and in vivo validation, including assessments of expression, immunogenicity (antibody and/or T-cell responses), protective efficacy in animal models, and safety/efficacy in human populations. Thus, RV is best viewed as a powerful filtering and prioritization tool that directs limited experimental resources toward the most promising candidates, thereby improving the overall efficiency of vaccine development.

The evolution from rule-based filtering methods (e.g., prioritizing surface proteins) to data-driven machine learning (ML) and deep learning (DL) reflects the field's recognition of the limitations of simplistic biological assumptions. ML models, such as Vaxign-ML, aim to learn complex patterns associated with protective immunity from empirical data, such as the Protegen database of experimentally validated protective antigens. While ML tools outperform traditional methods in benchmark tests, even the best models fall short of 100% accuracy, underscoring the probabilistic nature of RV predictions and the persistent challenge of precisely modeling intricate immunological processes in silico. This trajectory highlights ongoing efforts to improve predictive power while acknowledging inherent uncertainties, reinforcing that RV outputs are probabilistic rather than deterministic.

Ultimately, RV's primary value lies in its role as a prioritization tool, not a definitive predictor of clinical success. By integrating diverse bioinformatic evidence, it generates ranked lists of candidates that narrow the vast search space to a tractable subset with higher a priori success probability. This is particularly valuable given the high attrition rates in vaccine development, where lack of efficacy remains a major cause of clinical failure regardless of discovery method. We have detailed this discussion in **Page 23-25**.

References:

1. Handa T, Saha A, Narayanan A, Ronzier E, Kumar P, Singla J, Tomar S. *Structural Virology: Graham BS, Gilman MSA, McLellan JS. Structure-Based Vaccine Antigen Design. Annu Rev Med. 2019 Jan 27;70:91-104.*
2. Koebnik R, Locher KP, Van Gelder P. *Structure and function of bacterial outer membrane proteins: barrels in a nutshell. Mol Microbiol. 2000 Jul;37(2):239-53.*

3. Bae JE, Schurig GG, Toth TE. Mice immune responses to *Brucella abortus* heat shock proteins. Use of baculovirus recombinant-expressing whole insect cells, purified *Brucella abortus* recombinant proteins, and a vaccinia virus recombinant as immunogens. *Vet Microbiol.* 2002 Aug 25;88(2):189-202.
4. Ribeiro LA, Azevedo V, Le Loir Y, Oliveira SC, Dieye Y, Piard JC, Gruss A, Langella P. Production and targeting of the *Brucella abortus* antigen L7/L12 in *Lactococcus lactis*: a first step towards food-grade live vaccines against brucellosis. *Appl Environ Microbiol.* 2002 Feb;68(2):910-6.
5. Sette A, Rappuoli R. Reverse vaccinology: developing vaccines in the era of genomics. *Immunity.* 2010 Oct 29;33(4):530-41.

Some technical points:

Q8. *The authors should describe the dataset (distribution of antigen species) and data leakage between train and test: is there influenza sequences in both train and test,*

Response 8. We sincerely appreciate the reviewer's insightful comments regarding dataset composition and potential data leakage issues. In response to these concerns, we have now provided comprehensive documentation of species distribution for protective antigens in Supplementary Figure 1 (Taxonomic distribution of protective antigens across pathogen species) , which collectively demonstrates the extensive taxonomic diversity of pathogens represented in our dataset, encompassing viral, bacterial, and eukaryotic parasitic species.

Among bacterial antigens, the predominant sources were human pathogenic species, such as:

- *Escherichia coli*
- *Streptococcus pyogenes*
- *Staphylococcus aureus*
- *Mycobacterium tuberculosis*
- *Streptococcus pneumoniae*
- *Streptococcus agalactiae*
- *Brucella* spp.

Viral antigens were primarily derived from clinically significant human pathogens such as:

- *Influenza virus*
- *Human immunodeficiency virus*
- *Vaccinia virus*
- *Herpes simplex virus types 1 and 2*
- *Human papillomavirus*
- *Ebola virus*

For eukaryotic antigens, the most frequent sources were pathogenic parasites such as:

- *Plasmodium spp.*
- *Toxoplasma gondii*
- *Trypanosoma cruzi*
- *Leishmania donovani*

Supplementary Figure 1. Taxonomic distribution of protective antigens across pathogen species.

Visual representation of protective antigen source diversity, with viral, bacterial, and eukaryotic pathogens ranked by their respective antigen representations in the database.

Regarding data leakage prevention, we implemented stringent separation protocols during dataset partitioning. The training and test sets were meticulously divided at an 80:20 ratio with exclusion of identical antigen sequences between sets. Our evaluation metrics substantiate this rigorous separation approach, with test set performance accurately reflecting genuine predictive capability rather than memorization artifacts.

Q9. I didn't understand if/how authors mitigate the risk to suggest self-similar antigens to humans. You don't want to end-up with EBV sequences that are related to myelin...

Response 9. We sincerely appreciate the reviewer's insightful concern regarding the critical issue of potential autoimmunity risks from self-similar antigens. In conventional reverse vaccinology pipelines, a fundamental safety measure involves performing comprehensive sequence alignment between candidate antigens and the host proteome (primarily human, with additional consideration of mouse and swine proteomes in some tools) to systematically exclude sequences exhibiting significant homology. This essential bioinformatic filtering step prevents vaccine-induced immune responses from cross-reacting with host tissues. For instance, the Vaxign platform explicitly incorporates homology filtering options against human, mouse, and pig proteomes [1].

Our study implemented rigorous curation protocols for the positive dataset of protective antigens, which were exclusively derived from Proteogen and literature-reported in vivo validated antigens. This stringent selection process inherently excludes sequences demonstrating significant host homology. Consequently, our model's predictions based on sequence and structural feature extraction are inherently biased against identifying host-similar antigens. This is empirically validated in our case study, where all top 10 predicted monkeypox vaccine candidates exhibited low sequence homology to human proteins (Supplementary Data 6).

Among the top 10 candidate antigens, BLAST analysis against the UniProt human protein database revealed:

- 7 candidates showed no significant hits (E-value <1e-5)
- 3 candidates (B6R, A5L, and E8L) showed limited homology (36.5%, 34.8%, and 38.2% respectively)

Notably, while antigens Like B6R show some protein homology to humans, but it has already been validated as a target antigen in mRNA-1769 and BNT166 vaccines, having successfully completed immunogenicity and safety evaluations in non-human primates, with planned Phase 1/2 clinical trials [2, 3].

Supplementary Data 6: Homology of Mpox antigens against human proteome. Detailed BLAST comparison results between the top 10 predicted Mpox antigens and human proteins from UniProt database, including E-values, sequence coverage, and percentage identity for all significant hits.

References:

1. He Y, Xiang Z, Mobley HL. Vaxign: the first web-based vaccine design program for reverse vaccinology and applications for vaccine development. *J Biomed Biotechnol.* 2010;2010:297505.
2. Mucker EM, Freyn AW, Bixler SL, Cizmeci D, Atyeo C, Earl PL, Natarajan H, Santos G, Frey TR, Levin RH, Meni A, Arunkumar GA, Stadlbauer D, Jorquera PA, Bennett H, Johnson JC, Hardcastle K, Americo JL, Cotter CA, Koehler JW, Davis CI, Shamblin JD, Ostrowski K, Raymond JL, Ricks KM, Carfi A, Yu WH, Sullivan NJ, Moss B, Alter G, Hooper JW. Comparison of protection against mpox following mRNA or modified vaccinia Ankara vaccination in nonhuman primates. *Cell.* 2024 Oct 3;187(20):5540-5553.e10.
3. Zuiani A, Dulberger CL, De Silva NS, Marquette M, Lu YJ, Palowitch GM, Dokic A, Sanchez-Velazquez R, Schlatterer K, Sarkar S, Kar S, Chawla B, Galeev A, Lindemann C, Rothenberg DA, Diao H, Walls AC, Addona TA, Mensa F, Vogel AB, Stuart LM, van der Most R, Srouji JR, Türeci Ö, Gaynor RB, Şahin U, Poran A. A multivalent mRNA monkeypox virus vaccine (BNT166) protects mice and macaques from orthopoxvirus disease. *Cell.* 2024 Mar 14;187(6):1363-1373.e12.

Q10. I don't see the reason for down-sampling the negative dataset to only 1000 proteins while hundred of thousands may be available. Oversampling of the positives would be a better way to balance the dataset while keeping diversity of negatives.

Response 10. We sincerely appreciate the reviewer's insightful comments regarding the critical challenge of class imbalance in protective antigen prediction. As correctly noted, truly effective protective antigens (positive samples) represent only a minor fraction of pathogen proteomes compared to other proteins (potential negative samples). This biological reality inevitably leads to severe class imbalance in training datasets, which can significantly compromise the performance of standard machine learning algorithms. Such algorithms often exhibit bias toward the majority class (negatives), resulting in poor predictive performance for the minority class (positives).

In response to this concern, we have implemented a designed sampling strategy to better approximate real-world conditions while maintaining model performance. Our final curated dataset comprises 600 experimentally validated protective antigens (positives) and 6000 negative samples, spanning diverse pathogen types including viruses, bacteria, and parasites. This 1:10 ratio was empirically determined to optimally balance biological realism with computational efficiency. Following this dataset construction, we proceeded with the development and evaluation of our classification model, incorporating appropriate techniques to address the inherent class imbalance. These data as well as modifications have been detailed in the responses above.

Q11. Regarding the experimental setup, I understand that the mice can not be challenged by the real MPOX virus, and therefore testing reactivity to another virus is fine, and not seeing a strong phenotype might be due to the use of a different virus. The serum neutralization assay is fair. But the neutralization assay is never a proof of protection. I would question the general conclusion that the model predicts a “good antigen”. It could also be that any surface antigen of MPOX would equally induce an immune response and give the same experimental result irrespective of what the model predicted. I think a key result that would strengthen the model use is: taking an expressed surface antigen of MPOX that is predicted to be the least immunogenic and showing that the mice are not responsive to this antigen.

Response 11. We sincerely appreciate the reviewer's insightful comments regarding the experimental validation of our predicted antigens. While we acknowledge the limitations of using the surrogate mousepox virus (ECTV) model rather than authentic MPXV due to biosafety constraints, we would like

to emphasize that this represents an established alternative system for evaluating orthopoxvirus vaccine candidates [1, 2]. Crucially, sequence alignment confirmed exceptionally high conservation (>96% identity) between the novel MPXV antigen (G10R) and its ECTV homolog (Supplementary Fig.3), supporting the biological relevance of the observed immune responses.

Supplementary Figure 3. Sequence alignment of MPXV G10R antigen and its ECTV homolog. The alignment reveals 96.47% amino acid identity (Clustal Omega), with secondary structural elements (α -helices, β -sheets) annotated.

Following the model updates, we conducted predictive prioritization of Mpx candidate antigens and performed re-analysis integrated with experimental validation results (Figure 5). Notably, among three highly scored antigens (M1R, Score=0.9131; J3R, Score= 0.7671; G10R, Score=0.7648) whose proteins were successfully expressed and subjected to experimental validation, two demonstrated measurable protective efficacy (Figure 6). In contrast, the lower-scoring antigen B9R (score = 0.2083), despite demonstrating robust immunogenicity as a secretory protein capable of eliciting potent humoral immune

responses, failed to confer any protective effects, thereby validating the model's predictive capacity for identifying protective antigens. While these findings align with computational objectives, we acknowledge the current limitation in sample size and emphasize the necessity for expanded comparative validation across diverse Mpox antigenic proteins to further substantiate these observations.

[editorial note: panel redacted for third-party material]

Figure 5. Prediction of novel candidate antigens for Mpox. **a**, The complicated transmission cycle of the Mpox virus. **b**, Distribution table of antigen scores for Mpox. **c**, Expression and SDS- PAGE identification of candidate antigens. **d**, Protein structure modeling of candidate antigens, with color-coded annotations: brown = validated vaccine target controls, green = model-prioritized high-scoring candidates, black = low-scoring negative controls.

[editorial note: panel redacted for third-party material]

Figure 6. Immune evaluation of candidate antigens for Mpox. **a**, Schematic of the immunization assessment process. Specific pathogen-free (SPF) female BALB/c mice (6–8 weeks old, $n = 6$ biologically independent mice per group) were immunized intramuscularly with 100 μ L of vaccine formulations at day 0 and day 14. **b**, Specific total IgG antibody levels in mouse immune serum (mean \pm SEM). **c**, Specific IgG antibody subclasses in mouse immune serum (mean \pm SEM). **d**, Detection of neutralizing antibodies against ectromelia virus (mean \pm SEM). **e**, Protective efficacy in a lethal orthopoxvirus challenge. One-way analysis of variance (ANOVA) followed by Tukey’s post hoc test and unpaired t-test were used to analyze differences between groups. Data were considered statistically significant at $*p < 0.05$, $**p < 0.01$, $***p < 0.001$, and $****p < 0.0001$.

To further evaluate the protective potential of our lead candidate, G10R, we isolated G10R-specific monoclonal antibodies (mAbs) from immunized animals using hybridoma technology. These mAbs were characterized by their binding affinity to G10R via ELISA (Supplementary Figure 4a). Additionally, we assessed their neutralizing activity against orthopoxvirus *in vitro* (Supplementary Figure 4b). Among

the tested mAbs, 5C12 demonstrated potent neutralization, with an effective concentration (EC₅₀) of about 0.1 µg/mL in neutralization assays. This robust activity provides compelling evidence that G10R is a functional protective antigen capable of eliciting high-affinity, neutralizing antibodies. Prior studies have established a strong correlation between *in vitro* neutralization and *in vivo* protection in animal models for orthopoxviruses [1, 3].

Critically, this represents the first definitive identification of G10R as a bona fide protective antigen - demonstrating specific immunogenic properties rather than nonspecific immunological artifacts - thereby validating our model's predictive capacity for novel vaccine target discovery. Although we recognize that neutralization assays cannot establish absolute correlates of protection, the concordant demonstration of both high-affinity antigen binding and robust *in vitro* neutralizing activity provides compelling evidence for G10R's viability as a candidate vaccine immunogen.

These data, now partially summarized in the Discussion section (**Page 22, Line 480**), underscore the model's ability to prioritize antigens with functional protective potential rather than merely selecting surface-exposed proteins. We agree that future work should include direct MPXV challenge in non-human primates to confirm clinical relevance.

Supplementary Figure 4. G10R-specific monoclonal antibodies exhibit high-affinity binding and neutralizing activity against orthopoxviruses. a. Binding affinity of monoclonal antibodies to G10R antigen measured by ELISA. b. Neutralization of Vaccinia Virus by Monoclonal Antibodies. G10R-specific monoclonal antibodies (mAbs) were screened using hybridoma technology.

References:

1. Panchanathan V, Chaudhri G, Karupiah G. Correlates of protective immunity in poxvirus infection:

where does antibody stand? Immunol Cell Biol. 2008 Jan;86(1):80-6.

2. Yang Y, Zhao X, Li Y, Zai X, Wang X, Zhang Y, Wang X, Lv P, Zhang J, Hou L, Xu J, Chen W. Rational design of a single-component mRNA vaccine against orthopoxvirus and SARS-CoV-2. *Sci China Life Sci.* 2024 Jun;67(6):1311-1313.
 3. Yu H, Resch W, Moss B. Poxvirus structural biology for application to vaccine design. *Trends Immunol.* 2025 May 7:S1471-4906(25)00094-8.
-

We are grateful for your invaluable guidance throughout this process. Your expertise has been instrumental in shaping the study's scientific rigor and impact. We eagerly await your feedback on the revised manuscript and remain fully committed to addressing any additional suggestions to further elevate the quality of this work.

Point-by-Point Response to Reviewers' Comments

We thank the reviewers for their time and insightful comments. We have carefully addressed each point below and believe the resulting revisions have significantly improved the manuscript. Additionally, as per request, we have prepared and submitted a comprehensive “Source Data” Excel file containing all underlying numerical data for the figures in the main manuscript and supplementary information, ensuring full transparency and reproducibility of our findings. Furthermore, we have taken the initiative to redraw all figures in the manuscript, enhancing their clarity and visual appeal while maintaining scientific accuracy. All changes in the revised manuscript are highlighted using the track changes feature for clarity.

Response to Reviewer #3

Comment (on code availability): *I see the last classifier code from generated datasets, but I don't see the code to generate the structural embeddings, maybe the authors could also release this code?*

Response:

We thank Reviewer #3 for the continued support and this valuable suggestion. We agree that providing the code for generating structural embeddings is crucial for transparency and reproducibility. In response, we have updated our public repository to include the complete, documented source code for the Neighborhood-Enhanced Graph Convolutional Network (NEGCN) model used to generate these embeddings, along with detailed instructions for its use. We have updated the “Data and Code Availability” section in the manuscript to reflect this addition and provide a direct link to the repository (**Page 31, Line 728**). The code and data required to reproduce the results in this study are openly available via the anonymized repository: <https://anonymous.4open.science/r/PLGDL-602A>

Response to Reviewer #5

Comment: *The authors have addressed all the comments raised by Reviewer 1, and there are no further outstanding issues.*

Response:

We thank Reviewer #5 for the positive assessment and for confirming that the concerns raised in the previous round of review have been fully resolved.

Response to Reviewer #6

We sincerely thank Reviewer #6 for the thorough and critical evaluation. The points raised touch upon

fundamental challenges in the field of computational vaccinology. Engaging with these comments has allowed us to perform new analyses and expand our discussion, which we believe has improved the rigor and context of our study.

Comment 1: *The authors' answers to the reviewer comments are complete and exhaustive. They have made numerous changes to improve the study and the manuscript. However, the weakness of the study is evident in the authors' response to Reviewer 2's second comment, which weakness is also noted by Reviewer 1. The reliability of the derived models is undermined by the collection of the training and test sets and the validation of the results. The data collected from the Protegen database contain protective antigens proven through in vivo animal studies on viral, bacterial, and parasitic pathogens. The immune system's reaction to viral, bacterial, and parasitic infections follows different mechanisms. Combining antigens with such differences in immune mechanisms in training and test sets is inappropriate. For that reason, some existing tools that the authors compare their model against use separate models for pathogens with distinct origins.*

Response:

We fully agree with the reviewer's foundational premise that the downstream immune system pathways elicited by viral, bacterial, and eukaryotic pathogens are mechanistically distinct. For instance, intracellular viral infections are primarily cleared by cell-mediated immunity, whereas extracellular bacteria are typically targeted by humoral immunity (1). However, our model is not designed to predict the type of immune response an antigen will induce. Instead, the central objective of our PLGDL framework is to identify the intrinsic, learnable physicochemical and structural properties of a protein that make it a successful target for a protective immune response, regardless of the specific downstream mechanism.

This approach is grounded in the principles of structural vaccinology, which posits that protective antigens, even those from disparate pathogens, may share conserved structural motifs, topological features, or biophysical characteristics that are fundamental to their ability to elicit a protective response but are not readily apparent from primary sequence analysis alone. As stated in our manuscript, “Our research hypothesis stems from the intriguing observation that many successful vaccine target antigens, despite substantial sequence divergence, share common structural features”. For example, viral envelope glycoproteins (e.g., influenza HA, RSV F protein) often adopt trimeric conformations, while many bacterial outer membrane proteins exhibit conserved beta-barrel structures (2, 3). These structural commonalities suggest the existence of a learnable, pathogen-agnostic “grammar” of protective antigenicity. Our model's architecture, which synergistically combines a protein language model (to capture sequence-level context) and a geometric deep learning model (to capture 3D structural information), is explicitly designed to decipher this grammar. Therefore, the model is not learning the intricacies of host immunology; rather, it

is learning to recognize the biophysical patterns encoded in the stimulus (the protein antigen) that are correlated with the outcome of protection. The decision to use a unified dataset is thus a deliberate methodological choice to test this very hypothesis: that there are generalizable features of protective antigens that transcend pathogen-specific immune pathways.

To directly address the reviewer’s valid concern, we performed a new analysis to test whether training on a combined, diverse dataset hinders or helps performance on pathogen-specific tasks. We trained “specialist” models on pathogen-specific subsets of our data (i.e., a “Virus only” model, a “Bacteria only” model, and a “Eukaryota only” model). We then compared the performance of these specialist models against our original “generalist” PLGDL model (trained on the combined “Virus+Bacteria+Eukaryota” dataset) when evaluated on each pathogen-specific test set. The results, presented below, unequivocally demonstrate that the generalist model achieves superior or comparable performance across all pathogen classes.

Supplementary Table 3. Performance of PLGDL and its variants trained on pathogen-specific subsets

Test Set	Training Set	Accuracy	Precision	Recall	F1 Score	ROC-AUC	PR-AUC	MCC
Bacteria	Bacteria only	0.946	0.688	0.571	0.624	0.922	0.660	0.598
	Virus+Bacteria+Eukaryota	0.955	0.804	0.577	0.672	0.912	0.708	0.658
Eukaryota	Eukaryota only	0.942	0.688	0.579	0.629	0.940	0.685	0.600
	Virus+Bacteria+Eukaryota	0.954	0.769	0.588	0.667	0.893	0.718	0.649
Virus	Virus only	0.867	0.786	0.458	0.579	0.764	0.583	0.532
	Virus+Bacteria+Eukaryota	0.872	0.667	0.640	0.653	0.872	0.727	0.575

Analysis for Bacteria: The generalist model again outperforms the specialist model in 5 out of 7 metrics, including higher Precision (0.804 vs 0.688), Recall (0.577 vs 0.571), F1 Score (0.672 vs. 0.624), PR-AUC (0.708 vs. 0.660), and MCC (0.658 vs. 0.598).

Analysis for Eukaryota: While the specialist model shows slightly higher ROC-AUC, the generalist model demonstrates superior performance in metrics that are arguably more important for imbalanced classification, including F1 Score (0.667 vs. 0.629), and MCC (0.649 vs. 0.600).

Analysis for Viruses: The generalist model shows superior performance in 6 out of 7 metrics. Most

notably, there are substantial improvements in MCC (0.575 vs. 0.532) and PR-AUC (0.727 vs. 0.583), which are critical indicators of discriminative ability, especially on imbalanced datasets.

This new data provides empirical evidence that combining antigens from diverse pathogen origins does not, as the reviewer worried, create an “inappropriate” training set. On the contrary, it enables the model to learn more generalizable and robust features of protective antigenicity, leading to improved predictive performance. In summary, this new empirical evidence, demonstrates that our use of a unified dataset is not a weakness but a methodological strength that enhances the model's generalizability and predictive power. We have added this new analysis and an expanded discussion of these principles to the revised manuscript (**Page 10, Line 212; Page 19, Line 398; Page 27, Line 625**).

References:

1. Verhoef J, van Kessel K, Snippe H. Immune Response in Human Pathology: Infections Caused by Bacteria, Viruses, Fungi, and Parasites. Nijkamp and Parnham's Principles of Immunopharmacology. 2019 Feb 23:165–78.
2. Graham, B.S., Gilman, M.S.A., McLellan, J.S.: Structure-based vaccine antigen design. Annual Review of Medicine 70, 91–104 (2019)
3. Koebnik, R., Locher, K.P., Van Gelder, P.: Structure and function of bacterial outer membrane proteins: barrels in a nutshell. Molecular Microbiology 37(2), 239–253 (2000)

Comment 2: *Another issue is that all the antigens in the sets are protective in animals, but are not proven to be protective in humans. Using such sets to develop and evaluate models for predicting human protective antigens is problematic because of the differences between human and animal immune systems. Protective antigens proven in animals can be used to create predictive models, but for applicability to humans, the models must be carefully validated—and ideally fine-tuned—with human-derived data; otherwise, their predictions could be misleading. Hence, the comparisons with other tools and methods listed in Table 2 for predicting protective antigens are unreliable.*

Response:

We agree with the reviewer that animal models are not perfect surrogates for the human immune system and that preclinical data does not guarantee clinical success. This is a well-recognized, fundamental, and universal challenge for the entire field of vaccine development, not a unique flaw of our study. Differences in genetics, physiology, and pathogen tropism mean that no single animal model can perfectly recapitulate

human immune responses.

However, *in vivo* animal protection studies remain the indispensable “gold standard” for validating the protective efficacy of a vaccine candidate prior to human trials (1). A comprehensive database of “human protective antigens” does not and cannot exist, as its creation would require systematic, controlled pathogen challenge studies in human subjects, which is ethically prohibitive. In such cases where human efficacy studies are not ethical or feasible, regulatory bodies like the U.S. FDA have established pathways such as the "Animal Rule" that allow for approval based on well-controlled animal studies (2). Therefore, using the highest-quality available animal protection data is the standard, accepted, and most scientifically responsible approach for training and benchmarking computational tools in this domain. Our positive dataset was stringently curated from the Protegen database and PubMed, with a key inclusion criterion being demonstrated *in vivo* protective efficacy in a pathogen challenge model, ensuring our model is trained on the most rigorous ground-truth data currently available (3).

It is crucial to correctly position the PLGDL model within the broader vaccine development workflow. PLGDL is not intended to be a definitive predictor of clinical success in humans. Rather, its purpose is to serve as a powerful, high-throughput *in silico* screening and prioritization tool within the Reverse Vaccinology (RV) pipeline. The core value of such a tool lies in its ability to analyze an entire pathogen proteome—comprising hundreds or thousands of proteins—and identify a small, manageable list of the most promising candidates for expensive and labor-intensive experimental validation. This dramatically accelerates the discovery phase and, importantly, can reduce the overall reliance on animal testing by minimizing the number of non-viable candidates that enter the experimental pipeline, a goal actively promoted by regulatory agencies.

Our Mpox case study serves as a real-world demonstration of this utility. The PLGDL model rapidly screened over 190 Mpox proteins, correctly identified six known protective antigens within its top 10 predictions, and—most significantly—discovered the novel candidate G10R. Our subsequent experimental validation confirmed that G10R confers partial protection against a lethal orthopoxvirus challenge, thereby validating the model's real-world utility in identifying biologically relevant and effective vaccine candidates. The fact that our model, trained on a diverse set of historical animal-derived antigens, could pinpoint a functionally critical and conserved component of the viral entry machinery underscores that it is learning deep, biologically meaningful patterns directly relevant to protection, strongly supporting its utility.

The reviewer suggests that our comparisons with other tools in Table 2 are unreliable due to the use of animal data. However, the benchmarked tools (e.g., Vaxign-ML, Vaxign-DL) were themselves developed

and are routinely evaluated using the same types of animal-derived datasets (5,6). Therefore, our comparison is a appropriate evaluation of computational performance on the accepted data standard within the field. This analysis correctly establishes that PLGDL demonstrates superior predictive power relative to the current state-of-the-art on this benchmark.

We hope this explanation clarifies the model's intended role and validates its utility within the current constraints and realities of vaccine development. We have revised the Discussion section to better articulate these points (**Page 21, Line 465; Page 21, Line 472**).

References:

1. Chiarot E, Pizza M. Animal models in vaccinology: state of the art and future perspectives for an animal-free approach. *Curr Opin Microbiol.* 2022 Apr;66:46-55. doi: 10.1016/j.mib.2021.11.014. Epub 2021 Dec 22. PMID: 34953265.
2. Snoy PJ. Establishing efficacy of human products using animals: the US food and drug administration's "animal rule". *Vet Pathol.* 2010 Sep;47(5):774-8. doi: 10.1177/0300985810372506. Epub 2010 Jun 15. PMID: 20551476.
3. Yang, B., Sayers, S., Xiang, Z., He, Y.: Protegen: A web-based protective antigen database and analysis system. *Nucleic Acids Res.* 39(suppl 1), 1073–1078 (2011)
4. Doytchinova, I.A., Flower, D.R.: VaxiJen: A server for prediction of protective antigens, tumour antigens and subunit vaccines. *BMC Bioinformatics* 8, 1–7 (2007)
5. Ong, E., Wang, H., Wong, M.U., Seetharaman, M., Valdez, N., He, Y.: Vaxign- ML: Supervised machine learning reverse vaccinology model for improved prediction of bacterial protective antigens. *Bioinformatics* 36(10), 3185–3191 (2020)
6. Zhang, Y., Huffman, A., Johnson, J., He, Y.: Vaxign-DL: A deep learning-based method for vaccine design and its evaluation. *bioRxiv* (2023) <https://doi.org/10.1101/2023.11.29.569096>

We hope that these revisions, particularly the new cross-validation analysis, have addressed the reviewers' concerns. We believe the manuscript is now significantly improved and makes a timely and valuable contribution to the field.

Thank you again for the opportunity to revise our work.

Point-by-Point Response to Reviewer Comments

We extend our sincere gratitude to the reviewers for their time and insightful feedback throughout the review process, which has improved the clarity and impact of this manuscript.

Reviewer #3

Comment: I have no more requests.

Response: We thank Reviewer #3 for their positive assessment and for confirming that their previous concerns have been fully addressed.

Reviewer #6

Comment: The revision is thorough and well-reasoned. The new analyses convincingly address previous concerns about dataset composition and model validation, and the discussion now provides a clear methodological rationale. Before final acceptance, I recommend a few minor editorial adjustments:

(1) clarify in the Abstract that PLGDL is intended as a screening/prioritization tool, not a direct predictor of human efficacy;

(2) Reiterate this scope and limitation in the Discussion.

With these small refinements, the manuscript will be ready for publication.

Response: We thank Reviewer #6 for this positive and highly constructive feedback. We appreciate their assessment that our revision is “thorough and well-reasoned.” We agree completely with the recommended editorial adjustments to ensure the scope of our model is clearly and accurately articulated to readers. We have implemented both suggestions as follows:

Response to (1): We have revised the final sentence of the Abstract to explicitly state the model’s role as a prioritization tool.

The revised sentence in the Abstract now reads:

“Our study thus provides a high-performance tool for candidate antigen prioritization, synergistically leveraging protein language and geometric deep learning to facilitate rapid

vaccine development.”

Response to (2): We agree this is a critical point of clarification. We have added a dedicated paragraph to the Discussion section to explicitly state the model’s intended scope and its limitations, distinguishing it from a predictor of clinical success in humans.

The newly added text in the Discussion section reads:

“Given the intricate and dynamic nature of the immune system, forecasting the antigenicity of a vaccine or antigen is a formidable task. The PLGDL model is not intended to be a definitive predictor of clinical success in humans. Rather, its purpose is to serve as a powerful, high-throughput *in silico* screening and prioritization tool within the modern RV pipeline. The core value of such a tool lies in its ability to analyze an entire pathogen proteome—comprising hundreds or thousands of proteins—and identify a small, manageable list of the most promising candidates for expensive and labor-intensive experimental validation. As demonstrated by our Mpox case study, this dramatically accelerates the discovery phase.”

We believe these revisions fully address the reviewer’s insightful suggestions and appropriately contextualize our findings for the readership.